# TOPOLOGICAL GRAPH NEURAL NETWORKS

**Max Horn**[1,2,*]   **Edward De Brouwer**[3,*]   **Michael Moor**[1,2]   **Yves Moreau**[3]
**Bastian Rieck**[1,2,4,5,†]   **Karsten Borgwardt**[1,2,†]

[1]Department of Biosystems Science and Engineering, ETH Zurich, 4058 Basel, Switzerland
[2]SIB Swiss Institute of Bioinformatics, Switzerland
[3]ESAT-STADIUS, KU Leuven, 3001 Leuven, Belgium
[4]Institute of AI for Health, Helmholtz Munich, 85764 Neuherberg, Germany
[5]Technical University of Munich, 80333 Munich, Germany
[*]These authors contributed equally.
[†]These authors jointly supervised this work.

## ABSTRACT

Graph neural networks (GNNs) are a powerful architecture for tackling graph learning tasks, yet have been shown to be oblivious to eminent substructures such as cycles. We present TOGL, a novel *layer* that incorporates global topological information of a graph using persistent homology. TOGL can be easily integrated into *any type* of GNN and is strictly more expressive (in terms the Weisfeiler–Lehman graph isomorphism test) than message-passing GNNs. Augmenting GNNs with TOGL leads to improved predictive performance for graph and node classification tasks, both on synthetic data sets, which can be classified by humans using their topology but not by ordinary GNNs, and on real-world data.

## 1 INTRODUCTION

Graphs are a natural representation of structured data sets in many domains, including bioinformatics, image processing, and social network analysis. Numerous methods address the two dominant graph learning tasks of graph classification or node classification. In particular, *graph neural networks* (GNNs) describe a flexible set of architectures for such tasks and have seen many successful applications over recent years (Wu et al., 2021). At their core, many GNNs are based on iterative message passing schemes (see Shervashidze and Borgwardt (2009) for an introduction to iterative message passing in graphs and Sanchez-Lengeling et al. (2021) for an introduction to GNNs). Since these schemes are collating information over the neighbours of every node, GNNs cannot necessarily capture certain topological structures in graphs, such as cycles (Bouritsas et al., 2021). These structures are highly relevant for applications that require connectivity information, such as the analysis of molecular graphs (Hofer et al., 2020; Swenson et al., 2020).

We address this issue by proposing a Topological Graph Layer (TOGL) that can be easily integrated into any GNN to make it 'topology-aware.' Our method is rooted in the emerging field of topological data analysis (TDA), which focuses on describing coarse structures that can be used to study the shape of complex structured and unstructured data sets at multiple scales. We thus obtain a generic way to augment existing GNNs and increase their expressivity in graph learning tasks. Figure 1 provides a motivational example that showcases the potential benefits of using topological information: (i) high predictive performance is reached *earlier* for a *smaller* number of layers, and (ii) learnable topological representations outperform fixed ones if more complex topological structures are present in a data set.

**Our contributions.** We propose TOGL, a novel layer based on TDA concepts that can be integrated into any GNN. Our layer is differentiable and capable of learning contrasting topological representations of a graph. We prove that TOGL enhances expressivity of a GNN since it incorporates the ability to work with multi-scale topological information in a graph. Moreover, we show that TOGL improves predictive performance of several GNN architectures when topological information is relevant for the respective task.

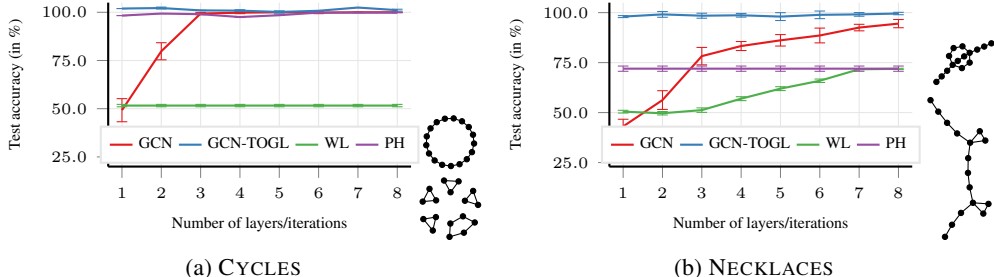

|  |  |
|:---:|:---:|
| (a) CYCLES | (b) NECKLACES |

Figure 1: As a motivating example, we introduce two topology-based data sets whose graphs can be easily distinguished by humans; the left data set can be trivially classified by all topology-based methods, while the right data set necessitates *learnable* topological features. We show the performance of (i) a GCN with $k$ layers, (ii) our layer TOGL (integrated into a GCN with $k - 1$ layers): GCN-TOGL, (iii) the Weisfeiler–Lehman (WL) graph kernel using vertex degrees as the node features, and (iv) a method based on static topological features (PH). Next to the performance charts, we display examples of graphs of each class for each of the data sets.

## 2 BACKGROUND: COMPUTATIONAL TOPOLOGY

We consider undirected graphs of the form $G = (V, E)$ with a set of vertices $V$ and a set of edges $E \subseteq V \times V$. The basic topological features of such a graph $G$ are the number of connected components $\beta_0$ and the number of cycles $\beta_1$. These counts are also known as the 0-dimensional and 1-dimensional *Betti numbers*, respectively; they are invariant under graph isomorphism (Hatcher, 2002, pp. 103–133) and can be computed efficiently. The expressivity of Betti numbers can be increased using a *graph filtration*, i.e. a sequence of nested subgraphs of $G$ such that $\emptyset = G^{(0)} \subseteq G^{(1)} \subseteq G^{(2)} \subseteq \cdots \subseteq G^{(n-1)} \subseteq G^{(n)} = G$. A filtration makes it possible to obtain more insights into the graph by 'monitoring' topological features of *each* $G^{(i)}$ and calculating their topological relevance, also referred to as their *persistence*. If a topological feature appears for the first time in $G^{(i)}$ and disappears in $G^{(j)}$, we assign this feature a persistence of $j - i$. Equivalently, we can represent the feature as a tuple $(i, j)$, which we collect in a *persistence diagram* $\mathcal{D}$. If a feature never disappears, we represent it by a tuple $(i, \infty)$; such features are the ones that are counted for the respective Betti numbers. This process was formalised and extended to a wider class of structured data sets, namely simplicial complexes, and is known under the name of *persistent homology*. One of its core concepts is the use of a filtration function $f : V \to \mathbb{R}^d$, with $d = 1$ typically, to accentuate certain structural features of a graph. This replaces the aforementioned tuples of the form $(i, j)$ by tuples based on the values of $f$, i.e. $(f_i, f_j)$. Persistent homology has shown excellent promise in different areas of machine learning research (see Hensel et al. (2021) for a recent survey and Appendix A for a more technical description of persistent homology), with existing work stressing that choosing or learning an appropriate filtration function $f$ is crucial for high predictive performance (Hofer et al., 2020; Zhao and Wang, 2019).

**Notation.** We denote the calculation of persistence diagrams of a graph $G$ under some filtration $f$ by $\mathbf{ph}(G, f)$. This will result in two persistence diagrams $\mathcal{D}^{(0)}, \mathcal{D}^{(1)}$, containing information about topological features in dimension 0 (connected components) and dimension 1 (cycles). The cardinality of $\mathcal{D}^{(0)}$ is equal to the number of nodes $n$ in the graphs and each tuple in the 0-dimensional diagram is associated with the *vertex* that created it. The cardinality of $\mathcal{D}^{(1)}$ is the number of cycles; we pair each tuple in $\mathcal{D}^{(1)}$ with the *edge* that created it. Unpaired edges—edges that are not used to create a cycle—are assigned a 'dummy' tuple value, such as $(0, 0)$. All other edges will be paired with the maximum value of the filtration, following previous work by Hofer et al. (2017).

## 3 RELATED WORK

Graph representation learning has received a large amount of attention by the machine learning community. *Graph kernel methods* address graph classification via (implicit or explicit) embeddings in Reproducing Kernel Hilbert Spaces (Borgwardt et al., 2020; Kriege et al., 2020; Nikolentzos et al., 2019). While powerful and expressive, they cannot capture partial similarities between neighbour-

hoods. This can be achieved by *graph neural networks*, which typically employ message passing on graphs to learn hidden representations of graph structures (Kipf and Welling, 2017; Wu et al., 2021). Recent work in this domain is abundant and includes attempts to utilise additional substructures (Bouritsas et al., 2021) as well as defining higher-order message passing schemes (Morris et al., 2019) or algorithms that generalise message passing to more complex domains (Bodnar et al., 2021).

Our approach falls into the realm of topological data analysis (Edelsbrunner and Harer, 2010) and employs *persistent homology*, a technique for calculating topological features—such as connected components and cycles—of structured data sets. These features are known to be highly characteristic, leading to successful topology-driven graph machine learning approaches (Hofer et al., 2017; 2020; Rieck et al., 2019; Zhao and Wang, 2019). At their core is the notion of a *filtration*, i.e. a sequence of nested subgraphs (or simplicial complexes in a higher-dimensional setting). Choosing the right filtration is known to be crucial for obtaining good performance (Zhao and Wang, 2019). This used to be a daunting task because persistent homology calculations are inherently discrete. Recent advances in proving differentiability enable proper end-to-end training of persistent homology (Carrière et al., 2021), thus opening the door for hybrid methods of increased expressivity by integrating the somewhat complementary view of topological features. Our method TOGL builds on the theoretical framework by Hofer et al. (2020), who (i) demonstrated that the *output* of a GNN can be used to 'learn' one task-specific filtration function, and (ii) described the conditions under which a filtration function $f$ is differentiable. This work culminated in GFL, a topological `readout` function that exhibited improved predictive performance for graph classification tasks. We substantially extend the utility of topological features by making existing GNNs 'topology-aware' through the development of a generic layer that makes topological information available to *all* downstream GNN layers: TOGL can be integrated into any GNN architecture, enabling the creation of hybrid models whose expressivity is provably more powerful than that of a GNN alone. Moreover, while GFL only uses the output of a GNN to drive the calculation of topological features by means of a single filtration (thus limiting the applicability of the approach, as the topological features cannot inform the remainder of a network), TOGL learns multiple filtrations of a graph in an end-to-end manner. More precisely, TOGL includes topological information in the hidden representations of nodes, enabling networks to change the importance of the topological signal. Closest to the scope of TOGL is Zhao et al. (2020), who enhanced GNNs using topological information for node classification. In their framework, however, topology is only used to provide additional scalar-valued weights for the message passing scheme, and topological features are only calculated over small neighbourhoods, making use of a static vectorisation technique of persistence diagrams. Similarly, Wong and Vong (2021) use static, i.e. non-learnable, topological features for 3D shape segmentation. By contrast, TOGL, being end-to-end differentiable, is more general and permits the calculation of topological features at all scales—including graph-level features—as well as an integration into arbitrary GNN architectures.

## 4 TOGL: A TOPOLOGICAL GRAPH LAYER

TOGL is a new type of graph neural network layer that is capable of utilising multi-scale topological information of input graphs. In this section, we give a brief overview of the components of this layer before discussing algorithmic details, theoretical expressivity, computational complexity, and limitations. Figure 2 presents an overview of our method (we show only a single graph being encoded, but in practice, the layer operates on *batches* of graphs).

The layer takes as input a graph $G = (V, E)$ equipped with a set of $n$ vertices $V$ and a set of edges $E$, along with a set of $d$-dimensional node attribute vectors $x^{(v)} \in \mathbb{R}^d$ for $v \in V$. These node attributes can either be node features of a data set or hidden representations learnt by some GNN. We employ a family of $k$ vertex filtration functions of the form $f_i \colon \mathbb{R}^d \to \mathbb{R}$ for $i = 1, \ldots, k$. Each filtration function $f_i$ can focus on different properties of the graph. The image of $f_i$ is finite and results in a set of node values $a_i^{(1)} < \cdots < a_i^{(n)}$ such that the graph $G$ is filtered according to $\emptyset = G_i^{(0)} \subseteq G_i^{(1)} \subseteq \cdots \subseteq G_i^{(n)} = G$, where $G_i^{(j)} = \left( V_i^{(j)}, E_i^{(j)} \right)$, with $V_i^{(j)} := \left\{ v \in V \mid f_i \left( x^{(v)} \right) \leq a_i^{(j)} \right\}$, and $E_i^{(j)} := \left\{ v, w \in E \mid \max \left\{ f_i \left( x^{(v)} \right), f_i \left( x^{(w)} \right) \right\} \leq a_i^{(j)} \right\}$. Given this filtration, we calculate a set of persistence diagrams, i.e. $\mathbf{ph}(G, f_i) = \left\{ \mathcal{D}_i^{(0)}, \ldots, \mathcal{D}_i^{(l)} \right\}$. We fix $l = 1$ (i.e. we are capturing connected components and cycles) to simplify our current implementation, but our layer can be

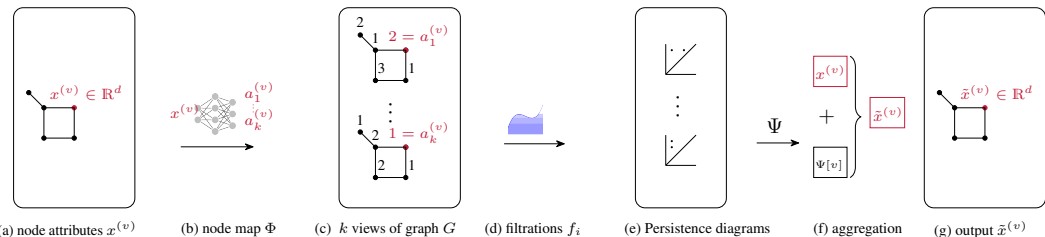

Figure 2: Overview of TOGL, our topological graph layer. a) The node attributes $x^{(v)} \in \mathbb{R}^d$ of graph $G$ serve as input. b) A network $\Phi$ maps $x^{(v)}$ to $k$ node values $\{a_1^{(v)}, \ldots, a_k^{(v)}\} \subset \mathbb{R}$. c) Applying $\Phi_k$ to the attributes of each vertex $v$ results in $k$ views of $G$. d) A vertex filtration $f_i$ is computed for the $i$th view of $G$. e) A set of $k$ persistence diagrams is encoded via the embedding function $\Psi$, where $\Psi[v]$ denotes the embedding of vertex $v$. f) The embedded topological features are combined with the input attributes $x^{(v)}$. g) Finally, this yields $\tilde{x}^{(v)} \in \mathbb{R}^d$, which acts as a new node representation augmented with multi-scale topological information.

extended to arbitrary values of $l$ (see Appendix C for a discussion). In order to benefit from representations that are trainable end-to-end, we use an *embedding function* $\Psi^{(l)} \colon \left\{ \mathcal{D}_1^{(l)}, \ldots, \mathcal{D}_k^{(l)} \right\} \to \mathbb{R}^{n' \times d}$ for embedding persistence diagrams into a high-dimensional space that will be used to obtain the vertex representations, where $n'$ is the number of *vertices* $n$ if $l = 0$ and the number of *edges* if $l = 1$. This step is crucial as it enables us to use the resulting topological features as node features, making TOGL a layer that can be integrated into arbitrary GNNs. We later explain the precise mapping of $\Psi^{(l)}$ from a set of diagrams to the elements of a graph.

**Details on filtration computation and output generation.** We compute our family of $k$ vertex-based filtrations using $\Phi \colon \mathbb{R}^d \to \mathbb{R}^k$, an MLP with a single hidden layer, such that $f_i := \pi_i \circ \Phi$, i.e. the projection of $\Phi$ to the $i$th dimension. We apply $\Phi$ to the hidden representations $x^{(v)}$ of all vertices in the graph. Moreover, we treat the resulting persistence diagrams in dimension 0 and 1 differently. For dimension 0, we have a bijective mapping of tuples in the persistence diagram to the vertices of the graph, which was previously exploited in topological representation learning (Moor et al., 2020). Therefore, we aggregate $\Psi^{(0)}$ with the original node attribute vector $x^{(v)}$ of the graph in a residual fashion, i.e. $\tilde{x}^{(v)} = x^{(v)} + \Psi^{(0)} \left( \mathcal{D}_1^{(0)}, \ldots, \mathcal{D}_k^{(0)} \right) [v]$, where $\Psi^{(0)}[v]$ denotes taking $v$th row of $\Psi^{(0)}$ (i.e the topological embedding of vertex $v$). The output of our layer for dimension 0 therefore results in a new representation $\tilde{x}^{(v)} \in \mathbb{R}^d$ for each vertex $v$, making it compatible with any subsequent (GNN) layers. By contrast, $\Psi^{(1)}$ is pooled into a graph-level representation, to be used in the final classification layer of a GNN. This is necessary because there is no bijective mapping to the vertices, but rather to edges. For stability reasons (Bendich et al., 2020), we consider it more useful to have this information available *only* on the graph level. For further details on the computational aspects, please refer to Section A.4.

**Complexity and limitations.** Persistent homology can be calculated efficiently for dimensions 0 and 1, having a worst-case complexity of $\mathcal{O}\left(m\alpha\left(m\right)\right)$ for a graph with $m$ sorted edges, where $\alpha(\cdot)$ is the extremely slow-growing inverse Ackermann function, which can be treated as constant for all intents and purposes. The calculation of **ph** is therefore dominated by the complexity of sorting all edges, i.e. $\mathcal{O}\left(m\log m\right)$, making our approach efficient and scalable. Higher-dimensional persistent homology calculations unfortunately do *not* scale well, having a worst-case complexity of $\mathcal{O}\left(m^d\right)$ for calculating $d$-dimensional topological features, which is why we restrict ourselves to $l = 1$ here. Our approach is therefore limited to connected components and cycles. Plus, our filtrations are incapable of assessing topological feature interactions; this would require learning *multifiltrations* (Carlsson et al., 2009), which do not afford a concise, efficient representation as the scalar-valued filtrations discussed in this paper. We therefore leave their treatment to future work.

## 4.1 Choosing an Embedding Function $\Psi$

The embedding function $\Psi$ influences the resulting representation of the persistence diagrams calculated by our approach. It is therefore crucial to pick a class of functions that are sufficiently

powerful to result in expressive representations of a persistence diagram $\mathcal{D}$. We consider multiple types of embedding functions $\Psi$, namely (i) a novel approach based on `DeepSets` (Zaheer et al., 2017), (ii) the *rational hat* function introduced by Hofer et al. (2019), as well as (iii) the triangle point transformation, (iv) the Gaussian point transformation, and (v) the line point transformation, with the last three transformations being introduced in Carrière et al. (2020). Except for the deep sets approach, all of these transformations are *local* in that they apply to a single point in a persistence diagram without taking the other points into account. These functions can therefore be decomposed as $\Psi^{(j)}\left(\mathcal{D}_1^{(j)}, \ldots, \mathcal{D}_k^{(j)}\right)[v] = \widetilde{\Psi}^{(j)}\left(\mathcal{D}_1^{(j)}[v], \ldots, \mathcal{D}_k^{(j)}[v]\right)$ for an index $v$. By contrast, our novel deep sets approach uses *all tuples* in the persistence diagrams to compute embeddings. In practice, we did not find a significant advantage of using any of the functions defined above; we thus treat them as a hyperparameter and refer to Appendix I for a detailed analysis.

## 4.2 DIFFERENTIABILITY & EXPRESSIVE POWER

The right choice of $\Psi$ will lead to a differentiable downstream representation. The map $\mathbf{ph}(\cdot)$ was shown to be differentiable (Gameiro et al., 2016; Hofer et al., 2020; Moor et al., 2020; Poulenard et al., 2018), provided the filtration satisfies injectivity at the vertices. We have the following theorem, whose proof is due to Hofer et al. (2020, Lemma 1).

**Theorem 1.** *Let $f_\theta$ be a vertex filtration function $f_\theta\colon V \to \mathbb{R}$ with continuous parameters $\theta$, and let $\Psi$ be a differentiable embedding function of unspecified dimensions. If the vertex function values of $f_\theta$ are distinct for a specific set of parameters $\theta'$, i.e. $f_\theta(v) \neq f_\theta(w)$ for $v \neq w$, then the map $\theta \mapsto \Psi\left(\mathbf{ph}(G, f_\theta)\right)$ is differentiable at $\theta'$.*

This theorem is the basis for TOGL, as it states that the filtrations, and thus the resulting 'views' on a graph $G$, can be trained end-to-end. While Hofer et al. (2020) describe this for a *single* filtration, their proof can be directly extended to multiple filtrations as used in our approach.

The expressive power of graph neural networks is well-studied (Chen et al., 2020b; Xu et al., 2019) and typically assessed via the iterative Weisfeiler–Lehman label refinement scheme, denoted as WL[1]. Given a graph with an initial set of vertex labels, WL[1] collects the labels of neighbouring vertices for each vertex in a multiset and 'hashes' them into a new label, using a perfect hashing scheme so that vertices/neighbourhoods with the same labels are hashed to the same value. This procedure is repeated and stops either when a maximum number of iterations has been reached or no more label updates happen. The result of each iteration $h$ of the algorithm for a graph $G$ is a feature vector $\phi_G^{(h)}$ that contains individual label counts. Originally conceived as a test for graph isomorphism (Weisfeiler and Lehman, 1968), WL[1] has been successfully used for graph classification (Shervashidze and Borgwardt, 2009). The test runs in polynomial time, but is known to fail to distinguish between certain graphs, i.e. there are non-isomorphic graphs $G$ and $G'$ that obtain the same labelling by WL[1] Fürer (2017). Surprisingly, Xu et al. (2019) showed that standard graph neural networks based on message passing are *no more powerful* than WL[1]. Higher-order refinement schemes, which pass information over tuples of nodes, for instance, can be defined (Maron et al., 2019; Morris et al., 2019). Some of these variants are strictly more powerful (they can distinguish between more classes of graphs) than WL[1] but also computationally more expensive.

To prove the expressivity of our method, we will show that (i) TOGL can distinguish all the graphs WL[1] can distinguish, and (ii) that there are graphs that WL[1] cannot distinguish but TOGL can. The higher expressivity of TOGL does not necessarily imply that our approach will perform generally better. In fact, WL[1] and, by extension, GNNs, are capable of identifying *almost all* non-isomorphic graphs (Babai et al., 1980). However, the difference in expressivity implies that TOGL can capture features that cannot be captured by GNNs, which can improve predictive performance if those features cannot be obtained otherwise. Since persistent homology is an isomorphism invariant, we first show that we distinguish the same graphs that WL[1] distinguishes. We do this by showing the existence of an injective filtration function $f$[1], thus ensuring differentiability according to Theorem 1.

**Theorem 2.** *Persistent homology is at least as expressive as WL[1], i.e. if the WL[1] label sequences for two graphs $G$ and $G'$ diverge, there exists an injective filtration $f$ such that the corresponding $0$-dimensional persistence diagrams $\mathcal{D}_0$ and $\mathcal{D}_0'$ are not equal.*

---

[1]We drop $\theta$ in this notation since we do not require $f$ to have a set of continuous parameters here; the subsequent theorem is thus covering a more general case than Theorem 1.

*Proof sketch.* We first assume the existence of a sequence of WL[1] labels and show how to construct a filtration function $f$ from this. While $f$ will result in persistence diagrams that are different, thus serving to distinguish $G$ and $G'$, it does not necessarily satisfy injectivity. We therefore show that there is an injective function $\tilde{f}$ that is arbitrarily close to $f$ and whose corresponding persistence diagrams $\widetilde{\mathcal{D}_0}, \widetilde{\mathcal{D}'_0}$ do *not* coincide. Please refer to Appendix B for the detailed version of the proof. $\square$

The preceding theorem proves the *existence* of such a filtration function. Due to the capability of GNNs to approximate the Weisfeiler–Lehman test (Xu et al., 2019) and the link between graph isomorphism testing and universal function approximation capabilities (Chen et al., 2019, Theorem 4), we can deduce that they are also able to approximate $f$ and $\tilde{f}$, respectively. Yet, this does *not* mean that we always end up learning $f$ or $\tilde{f}$. This result merely demonstrates that our layer can theoretically perform *at least as well as* WL[1] when it comes to distinguishing non-isomorphic graphs; this does not generally translate into better predictive performance, though. In practice, TOGL may learn other filtration functions; injective filtrations based on WL[1] are not necessarily optimal for a specific task (we depict some of the learnt filtrations in Appendix G).

To prove that our layer is more expressive than a GCN, we show that there are pairs of graphs $G, G'$ that cannot be distinguished by WL[1] but that *can* be distinguished by $\mathbf{ph}(\cdot)$ and by TOGL, respectively: let $G$ be a graph consisting of the disjoint union of two triangles, i.e. ◁▷, and let $G'$ be a graph consisting of a hexagon, i.e. ⬡. WL[1] will be unable to distinguish these two graphs because all multisets in every iteration will be the same. Persistent homology, by contrast, can distinguish $G$ from $G'$ using their Betti numbers. We have $\beta_0(G) = \beta_1(G) = 2$, because $G$ consists of two connected components and two cycles, whereas $\beta_0(G') = \beta_1(G') = 1$ as $G'$ only consists of one connected component and one cycle. The characteristics captured by persistent homology are therefore different from the ones captured by WL[1]. Together with Theorem 2, this example implies that persistent homology is *strictly* more powerful than WL[1] (see Appendix C for an extended expressivity analysis using higher-order topological features). The expressive power of TOGL hinges on the expressive power of the filtration—making it crucial that we can learn it.

## 5 EXPERIMENTS

We showcase the empirical performance of TOGL on a set of synthetic and real-world data sets, with a primary focus on assessing in which scenarios topology can enhance and improve learning on graph. Next to demonstrating improved predictive performance for synthetic and structure-based data sets (Section 5.2 and Section 5.3), we also compare TOGL with existing topology-based algorithms (Section 5.5).

### 5.1 EXPERIMENTAL SETUP

Following the setup of Dwivedi et al. (2020), we ran all experiments according to a consistent training setup and a limited parameter budget to encourage comparability between architectures. For further details, please refer to Appendix D. In all tables and graphs, we report the mean test accuracy along with the standard deviation computed over the different folds. All experiments were tracked (Biewald, 2020); experimental logs, reports, and code will be made publicly available.

**Baselines and comparison partners.** We compare our method to several GNN architectures from the literature, namely (i) Graph Convolutional Networks (Kipf and Welling, 2017, GCN), (ii) Graph Attention Networks (Veličković et al., 2018, GAT), (iii) Gated-GCN (Bresson and Laurent, 2017), (iv) Graph Isomorphism Networks (Xu et al., 2019, GIN), (v) the Weisfeiler–Lehman kernel (Shervashidze and Borgwardt, 2009, WL), and (vi) WL-OA (Kriege et al., 2016). The performance of these methods has been assessed in benchmarking papers (Borgwardt et al., 2020; Dwivedi et al., 2020; Morris et al., 2020), whose experimental conditions are comparable. We use the *same* folds and hyperparameters as in the corresponding benchmarking papers to ensure a fair comparison.

**TOGL setup.** We add our layer to existing GNN architectures, replacing the second layer by TOGL, respectively.[2] For instance, GCN-4 refers to a GCN with four layers, while GCN-3-TOGL-1 refers

---

[2] We investigate the implications of different layer placements in Appendix H and find that the best placement depends on the data set; placing the layer second is a compromise choice.

to method that has been made 'topology-aware' using TOGL. This setup ensures that both the original and the modified architecture have approximately the same number of parameters. We treat the choice of the embedding function $\Psi$ as a hyperparameter in our training for all subsequent experiments. Appendix I provides a comprehensive assessment of the difference between the `DeepSets` approach (which is capable of capturing *interactions* between tuples in a persistence diagram) and decomposed embedding functions, which do not account for interactions.

## 5.2 PERFORMANCE ON SYNTHETIC DATA SETS

As an illustrative example depicted in Figure 1, we use two synthetic balanced 2-class data sets of 1000 graphs each. In the CYCLES data set (Figure 1a, right), we generate either one large cycle (class 0) or multiple small ones (class 1). These graphs can be easily distinguished by *any* topological approach because they differ in the number of connected components and the number of cycles. For the NECKLACES data set (Figure 1b, right), we generate 'chains' of vertices with either two individual cycles, or a 'merged' one. Both classes have the same *number* of cycles, but the incorporation of additional connectivity information along their neighbourhoods makes them distinguishable for TOGL, since appropriate filtrations for the data can be learnt. All synthetic data sets use node features consisting of 3-dimensional vectors sampled from a Normal distribution. As PH and WL would consider all instances of graphs being distinct and thus remove any potential signal, we used the node degrees as features instead as commonly done (Borgwardt et al., 2020).

For this illustrative example, we integrate TOGL with a GCN. We find that TOGL performs well even *without* any GCN layers—thus providing another empirical example of improved expressivity. Moreover, we observe that standard GCNs require at least four layers (for CYCLES) or more (for NECKLACES) to approach the performance of TOGL. WL[1], by contrast, fails to classify CYCLES and still exhibits a performance gap to the GCN for NECKLACES, thus showcasing the benefits of having access to a learnable node representation. It also underscores the observations in Section 4.2: the higher expressive power of WL[1] as compared to a standard GCN does not necessarily translate to higher predictive performance. Appendix F provides an extended analysis with more configurations; we find that (i) both cycles and connected components are crucial for reaching high predictive performance, and (ii) the static variant is performing slightly better than the standard GCN, but *significantly worse* than TOGL, due to its very limited access to topological information. A simple static filtration (based on node degrees), which we denote by PH, only works for the CYCLES data set (this is a consequence of the simpler structure of that data set, which can distinguished by Betti number information already), whereas the more complex structure of the NECKLACES data necessitates learning a task-based filtration.

## 5.3 STRUCTURE-BASED GRAPH AND NODE CLASSIFICATION PERFORMANCE

A recent survey (Borgwardt et al., 2020) showed that the node features of benchmark graphs already carry substantial information, thus suppressing the signal carried by the graph structure itself to some extent. This motivated us to prepare a set of experiments in which we classify graphs *solely based on graph structure*. We achieve this by replacing all node labels and node features by random ones, thus leaving only structural/topological information for the classification task. In the case of the PATTERN dataset we use the original node features which are random by construction (Dwivedi et al., 2020).

Table 1 depicts the results for graph and node classification tasks on such graphs; we observe a *clear advantage* of TOGL over its comparison partners: making an existing GNN topology-aware via TOGL improves predictive performance in virtually all instances. In some cases, for instance when comparing GCN-4 to GCN-3-TOGL-1 on MNIST, the gains are substantial with an increase of more than 8%. This also transfers to other model architectures, where in most cases TOGL improves the performance across datasets. Solely the GIN model on PROTEINS and the GAT model on PATTERN decrease in performance when incorporating topological features. The deterioration of TOGL on PATTERNS is especially significant. Nevertheless, the inferior performance is in line with the low performance of GAT in general compared to the other methods we considered. In this context the lower performance of TOGL is not surprising as it relies on the backbone model for the construction of a filtration. This demonstrates the utility of TOGL in making additional structural information available to improve classification performance.

Table 1: Results for the structure-based experiments. We depict the test accuracy obtained on various benchmark data sets when only considering structural information (i.e. the network has access to *uninformative* node features). For MOLHIV, the ROC-AUC is reported. Graph classification results are shown on the left, while node classification results are shown on the right. We compare three architectures (GCN-4, GIN-4, GAT-4) with corresponding models where one layer is replaced with TOGL and highlight the winner of each comparison in **bold**.

| | *Graph classification* | | | | | *Node classification* |
|---|---|---|---|---|---|---|
| METHOD | DD | ENZYMES | MNIST | PROTEINS | MOLHIV | PATTERN |
| GCN-4 | 68.0±3.6 | 22.0±3.3 | 76.2±0.5 | 68.8±2.8 | 66.4±1.8 | 85.5±0.4 |
| GCN-3-TOGL-1 | **75.1±2.1** | **30.3±6.5** | **84.8±0.4** | **73.8±4.3** | **69.4±1.8** | **86.6±0.1** |
| GIN-4 | 75.6±2.8 | 21.3±6.5 | 83.4±0.9 | **74.6±3.1** | **68.7±0.9** | 84.8±0.0 |
| GIN-3-TOGL-1 | **76.2±2.4** | **23.7±6.9** | **84.4±1.1** | 73.9±4.9 | 65.1±6.2 | **86.7±0.1** |
| GAT-4 | 63.3±3.7 | 21.7±2.9 | 63.2±10.4 | 67.5±2.6 | 51.8±5.6 | **73.1±1.9** |
| GAT-3-TOGL-1 | **75.7±2.1** | **23.5±6.1** | **77.2±10.5** | **72.4±4.6** | **68.6±1.7** | 59.6±3.3 |

Table 2: Test accuracy on benchmark data sets (following standard practice, we report weighted accuracy on CLUSTER and PATTERN). Methods printed in black have been run in our setup, while methods printed in grey are cited from the literature, i.e. Dwivedi et al. (2020), Morris et al. (2020) for IMDB-B and REDDIT-B, and Borgwardt et al. (2020) for WL/WL-OA results. Graph classification results are shown on the left; node classification results are shown on the right. Following Table 1, we take existing architectures and replace their second layer by TOGL; we use *italics* to denote the winner of each comparison. A **bold** value indicates the overall winner of a column, i.e. a data set.

| | *Graph classification* | | | | | | | *Node classification* |
|---|---|---|---|---|---|---|---|---|
| METHOD | CIFAR-10 | DD | ENZYMES | MNIST | PROTEINS-FULL | IMDB-B | REDDIT-B | CLUSTER |
| GATED-GCN-4 | **67.3±0.3** | 72.9±2.1 | 65.7±4.9 | **97.3±0.1** | **76.4±2.9** | — | — | 60.4±0.4 |
| WL | — | 77.7±2.0 | 54.3±0.9 | — | 73.1±0.5 | 71.2±0.5 | 78.0±0.6 | — |
| WL-OA | — | **77.8±1.2** | 58.9±0.9 | — | 73.5±0.9 | 74.0±0.7 | 87.6±0.3 | — |
| GCN-4 | 54.2±1.5 | 72.8±4.1 | **65.8±4.6** | 90.0±0.3 | *76.1±2.4* | 68.6±4.9 | **92.8±1.7** | 57.0±0.9 |
| GCN-3-TOGL-1 | *61.7±1.0* | *73.2±4.7* | 53.0±9.2 | *95.5±0.2* | 76.0±3.9 | *72.0±2.3* | 89.4±2.2 | *60.4±0.2* |
| GIN-4 | 54.8±1.4 | 70.8±3.8 | *50.0±12.3* | *96.1±0.3* | 72.3±3.3 | 72.8±2.5 | 81.7±6.9 | 58.5±0.1 |
| GIN-3-TOGL-1 | *61.3±0.4* | *75.2±4.2* | 43.8±7.9 | 96.1±0.1 | *73.6±4.8* | **74.2±4.2** | 89.7±2.5 | *60.4±0.2* |
| GAT-4 | 57.4±0.6 | 71.1±3.1 | 26.8±4.1 | 94.1±0.3 | 71.3±5.4 | *73.2±4.1* | 44.2±6.6 | 56.6±0.4 |
| GAT-3-TOGL-1 | *63.9±1.2* | *73.7±2.9* | *51.5±7.3* | *95.9±0.3* | *75.2±3.9* | 70.8±8.0 | *89.5±8.7* | *58.4±3.7* |

Varying the number of layers in this experiment underscores the benefits of including TOGL in a standard GCN architecture. Specifically, an architecture that incorporates TOGL reaches high predictive performance earlier, i.e. with fewer layers, than other methods, thus reducing the risk of oversmoothing (Chen et al., 2020a). Figure 3 depicts this for MNIST; we observe similar effects on the other data sets. Appendix I presents a detailed performance comparison ,and extended analysis, and a discussion of the effects of different choices for an embedding function Ψ.

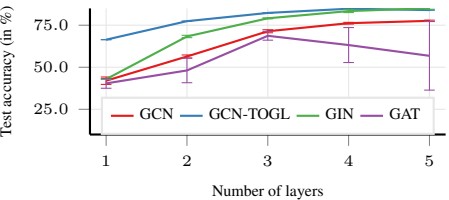

Figure 3: Classification performance when analysing the structural variant of MNIST.

## 5.4 PERFORMANCE ON BENCHMARK DATA SETS

Having ascertained the utility of TOGL for classifying data sets with topological information, we now analyse the effects of TOGL in standard graph and node classification tasks. Table 2 depicts the results on well-known benchmark data sets for graph and node classification. We see that TOGL performs better than its comparison partners (i.e. GCN-4, GIN-4, GAT-4) on most of the data sets, showcasing the benefits of substituting a layer of GNN with TOGL. Concerning ENZYMES performance, we experienced a severe degree of overfitting during training. This was exacerbated

Table 3: Test accuracy when comparing TOGL (integrated into a simplified architecture) with existing topology-based embedding functions or `readout` functions. Results shown in grey are cited from the respective papers (Carrière et al., 2020; Hofer et al., 2020). For GFL, we cite degree-based results so that its performance values pertain to the same scenario.

| METHOD | REDDIT-5K | IMDB-MULTI | NCI1 | REDDIT-B | IMDB-B |
|---|---|---|---|---|---|
| GFL | 55.7±2.1 | 49.7±2.9 | 71.2±2.1 | 90.2±2.8 | **74.5±4.6** |
| PersLay | 55.6 | 48.8 | 73.5 | — | 71.2 |
| GCN-1-TOGL-1 | **56.1±1.8** | **52.0±4.0** | **75.8±1.8** | 90.1±0.8 | 74.3±3.6 |
| GCN-1-TOGL-1 (static) | 55.5±1.8 | 48.3±4.9 | 75.1±1.2 | **90.4±1.4** | 72.2±2.1 |

by the fact that ENZYMES is the smallest of the compared data sets and we eschewed the tuning of regularisation hyperparameters such as 'dropout' for the sake of being comparable with the benchmark results (Dwivedi et al., 2020). With the GAT-based GNN, we generally observe an high degree of variance, as already reported in previous studies (Dwivedi et al., 2020). In particular, we experienced issues in training it on the REDDIT-B dataset. The addition of TOGL seems to address this issue, which we consider to underline the overall potential of topological features (at least for data sets with distinct topological information).

## 5.5 COMPARISON TO OTHER TOPOLOGY-BASED ALGORITHMS

In light of TOGL containing existing embedding functions $\Psi$ (Carrière et al., 2020), we compare its performance to other topology-based algorithms that have been used for graph classification. Table 3 summarises the performance (for comparison purposes, we also show the results for a 'static' variant of our layer; see Appendix D for more details). In order to permit a fair comparison, we integrate TOGL into a simpler GNN architecture, consisting of a single GCN message passing layer.

We observe that our filtration learning approach outperforms the fixed filtrations used by Carrière et al. (2020), highlighting the utility of a layer specifically designed to be integrated into GNNs. TOGL also fares well in comparison to the `readout` function by Hofer et al. (2020). While large standard deviations (due to the small size of the data sets) preclude an assessment of significant differences, we hypothesise that the benefits of having access to *multiple filtrations* will be more pronounced for large data sets containing more pronounced topological features, such as molecular graphs. Notice that in contrast to Hofer et al. (2020), our layer can be included at different stages of the overal GNN architecture. We investigate the impact of different positions in Appendix H and show that different positions can lead significantly different performance. These results further motivate the importance of a flexible topological *layer* as opposed to `readout` functions or static filtrations.

## 6 CONCLUSION

We presented TOGL, a generically-applicable layer that incorporates topological information into any GNN architecture. We proved that TOGL, due to its filtration functions (i.e. input functions) being learnable, is more expressive than WL[1], the Weisfeiler–Lehman test for graph isomorphism, and therefore also increases expressivity of GNNs. On data sets with pronounced topological structures, we found that our method helps GNNs obtain substantial gains in predictive performance. We also saw that the choice of function for embedding topological descriptors is crucial, with embedding functions that can handle interactions between individual topological features typically performing better than those that cannot. On benchmark data sets, we observed that our topology-aware approach can help improve predictive performance while overall exhibiting favourable performance in comparison to the literature. We also observed that topological information may sometimes lead to overfitting issues on smaller data sets, and leave the investigation of additional regularisation strategies for our method for future work. Furthermore, we hypothesise that the use of different filtration types (Milosavljević et al., 2011), together with advanced persistent homology algorithms, such as extended persistence (Cohen-Steiner et al., 2009), will prove beneficial for predictive performance.

## Reproducibility Statement

We have provided the code for our experiments along with the seeds used for training. All experiments were run on single GPUs, which avoids additional sources of randomness. Details are provided in Section 5.1. The parameters and performance metrics during training were tracked and will be made publicly available (Biewald, 2020). Our code is released under a BSD-3-Clause License and can be accessed under `https://github.com/BorgwardtLab/TOGL`.

## Acknowledgements

The authors would like to thank "Weights and Biases, Inc." for providing us with a free academic team account. This project was supported by the grant #2017-110 of the Strategic Focal Area "Personalized Health and Related Technologies (PHRT)" of the ETH Domain for the SPHN/PHRT Driver Project "Personalized Swiss Sepsis Study" and the Alfried Krupp Prize for Young University Teachers of the Alfried Krupp von Bohlen und Halbach-Stiftung (K.B.) Edward De Brouwer gratefully acknowledges support by an FWO-SB grant and support from NVIDIA for GPUs. Yves Moreau is funded by (1) Research Council KU Leuven: C14/18/092 SymBioSys3; CELSA-HIDUCTION, (2) Innovative Medicines Initiative: MELLODY, (3) Flemish Government (ELIXIR Belgium, IWT, FWO 06260) and (4) Impulsfonds AI: VR 2019 2203 DOC.0318/1QUATER Kenniscentrum Data en Maatschappij. Some computational resources and services used in this work were provided by the VSC (Flemish Supercomputer Center), funded by the Research Foundation - Flanders (FWO) and the Flemish Government – department EWI.

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

## A  TOPOLOGICAL DATA ANALYSIS

We provide a more formal introduction to persistent homology, the technique on which TOGL is fundamentally based. Persistent homology arose as one of the flagship approaches in the field of computational topology, which aims to make methods from this highly abstract branch of mathematics available for data analysis purposes.

To gain a better understanding, we will briefly take a panoramic tour through algebraic topology, starting from *simplicial homology*, an algebraic technique for 'calculating' the connectivity of topological spaces, represented in the form of simplicial complexes, i.e. generalised graphs. Simplicial homology is said to assess the connectivity of a topological space by 'counting its high-dimensional holes'. We will see how to make this description more precise.

### A.1  SIMPLICIAL HOMOLOGY

Simplicial complexes are the central concept in algebraic topology. A simplicial complex K consists of a set of *simplices* of certain dimensions, such as vertices (dimension 0), edges (dimension 1), and triangles (dimension 2). Each simplex $\sigma \in$ K has a set of faces, and each face $\tau$ has to satisfy $\tau \in$ K. Moreover, if $\sigma \cap \sigma' \neq \emptyset$ for $\sigma, \sigma' \in$ K, then $\sigma \cap \sigma' \in$ K. Thus, K is 'closed under calculating the faces of a simplex'. A graph can be seen as a low-dimensional simplicial complex that only contains 0-simplices (vertices) and 1-simplices (edges), so everything we say applies, *mutatis mutandis*, also to graphs.

**Chain groups.**  For a simplicial complex K, we denote by $C_d(K)$ the vector space generated over $\mathbb{Z}_2$ (the field with two elements). The elements of $C_d(K)$ are the $d$-simplices in K, or rather their *formal sums* with coefficients in $\mathbb{Z}_2$. For example, $\sigma + \tau$ is an element of the chain group, also called a *simplicial chain*. Addition is well-defined and easy to implement as an algorithm since a simplex can only be present or absent over $\mathbb{Z}_2$ coefficients.[3] The use of chain groups lies in providing the underlying vector space to formalise boundary calculations over a simplicial complex. The boundary calculations, in turn, are necessary to quantify the connectivity!

**Boundary homomorphism.**  Given a $d$-simplex $\sigma = (v_0, \ldots, v_d) \in$ K, we can formalise its face or 'boundary' calculation by defining the boundary operator $\partial_d \colon C_d(K) \to C_{d-1}(K)$ as a sum of the form

$$\partial_d(\sigma) := \sum_{i=0}^{d} (v_0, \ldots, v_{i-1}, v_{i+1}, \ldots, v_d), \tag{1}$$

i.e. we leave out every vertex $v_i$ of the simplex once. Since only sums are involved, this operator is seen to be a homomorphism between the chain groups; the calculation extends to $C_d(K)$ by linearity. The boundary homomorphism gives us a way to precisely define what we understand by connectivity. To this end, note that its *kernel* and *image* are well-defined. The kernel $\ker \partial_d$ contains all $d$-dimensional simplicial chains that do not have a boundary. We can make this more precise by using a construction from group theory.

**Homology groups.**  The last ingredient for the connectivity analysis involves calculating a special group, the homology group. The $d$th homology group $H_d(K)$ of K is defined as the *quotient group* $H_d(K) := \ker \partial_d / \operatorname{im} \partial_{d+1}$. The quotient operation can be thought of as calculating a special subset—the kernel of the boundary operator—and then *removing* another subset, namely the image of the boundary operator with an increased dimension. This behoves a short explanation. The main reason behind this operation is that *just* the kernel calculation is insufficient to properly count a hole. For example, if we take the three edges of any triangle, their boundary will always be empty, i.e. they are a part of $\ker \partial_1$. However, if the *interior* of the triangle is also part of the simplicial complex—in other words, if we have the corresponding 2-simplex as well—we should *not* count the edges as a hole. This is why we need to remove all elements in the image of $\partial_2$. Coincidentally, this also explains why cycles in a graph never 'vanish'—there are simply no 2-simplices available since the graph is only a 1-dimensional simplicial complex.

---

[3]Different choices of coefficient fields would be possible, but are rarely used for data analysis purposes.

**Betti numbers.** To fully close the loop, as it were, it turns out that we can calculate Betti numbers from homology groups. Specifically, the *rank* of the $d$th homology group—in the group-theoretical sense—is precisely the $d$th Betti number $\beta_d$, i.e. $\beta_d(\mathrm{K}) := \mathrm{rank}\, H_d(\mathrm{K})$. The sequence of Betti numbers $\beta_0, \ldots, \beta_d$ of a $d$-dimensional simplicial complex is commonly used as a complexity measure, and they can be used to discriminate manifolds. For example, a 2-sphere has Betti numbers $(1, 0, 1)$, while a 2-torus has Betti numbers $(1, 2, 1)$. As we outlined in the main text, Betti numbers are of limited use when dealing with complex graphs, however, because they are very coarse counts of features. It was this limited expressivity that prompted the development of persistent homology.

## A.2 PERSISTENT HOMOLOGY

Persistent homology is an extension of simplicial homology, which employs *filtrations* to imbue a simplicial complex K with scale information. Let us assume the existence of a function $f \colon \mathrm{K} \to \mathbb{R}$, which only attains a finite number of function values $f^{(0)} \le f^{(1)} \le \cdots \le \ldots f^{(m-1)} \le f^{(m)}$. We may now, as in the main text, sort K according to $f$, leading again to a nested sequence of simplicial complexes

$$\emptyset = \mathrm{K}^{(0)} \subseteq \mathrm{K}^{(1)} \subseteq \cdots \subseteq \mathrm{K}^{(m-1)} \subseteq \mathrm{K}^{(m)} = \mathrm{K}, \tag{2}$$

in which $\mathrm{K}^{(i)} := \left\{ \sigma \in K \mid f(\sigma) \le f^{(i)} \right\}$, i.e. each subset contains only those simplices whose function value is less than or equal to the threshold. In contrast to simplicial homology, the filtration holds potentially more information because it can track *changes*! For example, a topological feature might be *created* (a new connected component might arise) or *destroyed* (two connected components might merge into one), as we go from some $\mathrm{K}^{(i)}$ to $\mathrm{K}^{(i+1)}$. At its core, persistent homology is 'just' a way of tracking topological features, representing each one by a creation and destruction value $(f^{(i)}, f^{(j)}) \in \mathbb{R}^2$, with $i \le j$. In case a topological feature is still present in $\mathrm{K}^{(m)} = \mathrm{K}$, such as a cycle in a graph, it can also be assigned a tuple of the form $(f^{(i)}, \infty)$. Such tuples constitute *essential features* of a simplicial complex and are usually assigned a large destruction value or treated separately in an algorithm (Hofer et al., 2017). While it is also possible to obtain only tuples with finite persistence values, a process known as *extended persistence* (Cohen-Steiner et al., 2009), we focus only on 'ordinary' persistence in this paper because of the lower computational complexity.

**Persistent homology groups.** The filtration above is connected by the inclusion homomorphism between $\mathrm{K}^{(i)} \subseteq \mathrm{K}^{(i+1)}$. The respective boundary homomorphisms induce a homomorphism between corresponding homology groups of the filtration, i.e. a map $\mathrm{i}_d^{(i,j)} \colon H_d(\mathrm{K}_i) \to H_d(\mathrm{K}_j)$. This family of homomorphisms gives rise to a sequence of homology groups

$$
\begin{aligned}
H_d\left(\mathrm{K}^{(0)}\right) \xrightarrow{\mathrm{i}_d^{(0,1)}} H_d\left(\mathrm{K}^{(1)}\right) \xrightarrow{\mathrm{i}_d^{(1,2)}} \ldots \xrightarrow{\mathrm{i}_d^{(m-2,m-1)}} \\
H_d\left(\mathrm{K}^{(m-1)}\right) \xrightarrow{\mathrm{i}_d^{(m-1,m)}} H_d\left(\mathrm{K}^{(m)}\right) = H_d\left(\mathrm{K}\right)
\end{aligned} \tag{3}
$$

for every dimension $d$. For $i \le j$, the $d$th persistent homology group is defined as

$$H_d^{(i,j)} := \ker \partial_d\left(\mathrm{K}^{(i)}\right) / \left(\mathrm{im}\, \partial_{d+1}\left(\mathrm{K}^{(j)}\right) \cap \ker \partial_d\left(\mathrm{K}^{(i)}\right)\right). \tag{4}$$

Intuitively, this group contains all homology classes *created* in $\mathrm{K}^{(i)}$ that are *also* present in $\mathrm{K}^{(j)}$. Similar to the 'ordinary' Betti number, we may now define the $d$th persistent Betti number as the rank of this group, i.e.

$$\beta_d^{(i,j)} := \mathrm{rank}\, H_d^{(i,j)}. \tag{5}$$

Noting the set of indices $i, j$, we can see persistent homology as generating a *sequence* of Betti numbers, as opposed to just calculating a single number. This makes it possible for us to describe topological features in more detail, and summarise them in a *persistence diagram*.

**Persistence diagrams.** Given a filtration induced by a function $f \colon \mathrm{K} \to \mathbb{R}$ as described above, we store each tuple $(f^{(i)}, f^{(j)})$ with multiplicity

$$\mu_{i,j}^{(d)} := \left(\beta_d^{(i,j-1)} - \beta_d^{(i,j)}\right) - \left(\beta_d^{(i-1,j-1)} - \beta_d^{(i-1,j)}\right) \tag{6}$$

in the $d$th persistence diagram $\mathcal{D}_d$. Notice that for most pairs of indices, $\mu_{i,j}^{(d)} = 0$. Given a point $(x, y) \in \mathcal{D}_d$, we refer to the quantity $\mathrm{pers}(x, y) := |y - x|$ as its *persistence*. The idea of persistence arose in multiple contexts (Barannikov, 1994; Edelsbrunner et al., 2002; Verri et al., 1993), but it is nowadays commonly used to analyse functions on manifolds, where high persistence is seen to correspond to *features* of the function, while low persistence is typically considered *noise*.

## A.3 EXAMPLES OF GRAPH FILTRATIONS

Before discussing the computational complexity of persistent homology, we briefly provide some worked examples that highlight the impact of choosing different filtrations for analysing graphs. Filtrations are most conveniently thought of as arising from a function $f \colon G \to \mathbb{R}$, which assigns a scalar-valued function value to each node and edge of the graph by means of setting $f(u, v) := \max\{f(u), f(v)\}$ for an edge $(u, v)$. In this context, $f$ is often picked to measure certain salient vertex features of $G$, such as the degree (Hofer et al., 2017), or its structural role in terms of a heat kernel (Carrière et al., 2020). The resulting topological features are then assigned the respective function values, i.e. $(i, j) \mapsto (f(u_i), f(u_j))$. As a brief example, consider a degree-based filtration of a simple graph. The filtration values are shown as numbers next to a vertex; we use $a^{(j)}$ to denote the filtration value at step $j$ (this notation will be useful later when dealing with multiple filtrations). In each filtration step, the new nodes and edges added to the graph are shown in red, while black elements indicate the structure of the graph that already exists at this step.

Since all edges are inserted at $a^{(3)} = 3$, we obtain the following 0-dimensional persistence diagram $\mathcal{D}_0 = \{(1, \infty), (1, 3), (2, 3), (3, 3), (3, 3)\}$. The existence of a single tuple of the form $(\cdot, \infty)$ indicates that $\beta_0 = 1$. Similarly, there is only one cycle in the data set, which is created at $a^{(3)}$, leading to $\mathcal{D}_1 = \{(3, \infty)\}$ and $\beta_1 = 1$. If we change the filtration such that each vertex has a *unique* filtration value, we obtain a different ordering and different persistence tuples, as well as more filtration steps:

Here, connected components are not all created 'at once', but more gradually, leading to $\mathcal{D}_0 = \{(1, \infty), (3, 3), (2, 4), (4, 4), (5, 5)\}$. Of particular interest is the tuple $(2, 4)$. It was created by the vertex with filtration value 2. In $G^{(3)}$ it merges with another connected component, namely the one created by the vertex with function value 3. This leads to the tuple $(3, 3)$, because in each merge, we merge from the 'younger' component (the one arising *later* in the filtration) to the 'older' component (the arising *earlier* in the filtration). Again, there is only a single cycle in the data set, which we detect at $a^{(5)} = 5$, leading to $\mathcal{D}_1 = \{(5, \infty)\}$.

## A.4 COMPUTATIONAL DETAILS

The cardinality of the persistence diagram of dimension-0, $\mathcal{D}_0$ is equal to the number of vertices, $n$ in the graph. A natural pairing of persistence tuples then consists in assigning each tuple to the node that generated it. As for dimension-1, $\mathcal{D}_1$ contains as many tuples as cycles in the graph. However, there is no bijective mapping between dimension-1 persistence tuples and vertices. Rather, we link each dimension-1 tuple to the edge that created that particular cycle. To account for multiple distinct filtrations and because the same cycle can be assigned to different edges depending on the specific filtration function, we define a *dummy* tuple for edges that are not linked to any cycle for a particular filtration. The set of persistence diagrams $(\mathcal{D}_1^{(l)}, \ldots, \mathcal{D}_k^{(l)})$ can then be concatenated as a matrix in $\mathbb{R}^{d_l \times 2 \cdot k}$, with $d_l = |V|$ (the number of vertices in graph $G$) if $l = 0$ and $d_l = |E|$ (the number of

edges in $G$) if $l = 1$. Remarkably, that leads to $\Psi^{(l)}\left(\mathcal{D}_1^{(l)}, \ldots, \mathcal{D}_k^{(l)}\right)$ being an operator on a matrix, significantly facilitating computations.

Because of its natural indexing to the vertices, $\Psi^{(0)}\left(\mathcal{D}_1^{(0)}, \ldots, \mathcal{D}_k^{(0)}\right)$ can be mapped back to the graph as explained in Section 4. For $l = 1$, we pool $\Psi^{(1)}\left(\mathcal{D}_1^{(1)}, \ldots, \mathcal{D}_k^{(1)}\right)$ to a graph-level embedding, and mask out the edge indices that are not assigned a persistence pair in any of the $\mathcal{D}_k^{(1)}$.

## B  THEORETICAL EXPRESSIVITY OF TOGL: PROOFS

This section provides more details concerning the proofs about the expressivity of our method. We first state a 'weak' variant of our theorem, in which injectivity of the filtration function is not guaranteed. Next, we show that the filtration function $f$ constructed in this theorem can be used to prove the existence of an injective function $\tilde{f}$ that is arbitrarily close to $f$ and provides the same capabilities of distinguishing graphs.

**Theorem 3** (Expressivity, weak version). *Persistent homology is* at least *as expressive as WL[1], i.e. if the WL[1] label sequences for two graphs $G$ and $G'$ diverge, there is a filtration $f$ such that the corresponding $0$-dimensional persistence diagrams $\mathcal{D}_0$ and $\mathcal{D}_0'$ are not equal.*

*Proof.* Assume that the label sequences of $G$ and $G'$ diverge at iteration $h$. Thus, $\phi_G^{(h)} \neq \phi_{G'}^{(h)}$ and there exists at least one label whose count is different. Let $\mathcal{L}^{(h)} := \{l_1, l_2, \ldots\}$ be an enumeration of the finitely many hashed labels at iteration $h$. We can build a filtration function $f$ by assigning a vertex $v$ with label $l_i$ to its index, i.e. $f(v) := i$, and setting $f(v, w) := \max\{f(v), f(w)\}$ for an edge $(v, w)$. The resulting $0$-dimensional persistence diagram $\mathcal{D}_0$ (and $\mathcal{D}_0'$ for $G'$) will contain tuples of the form $(i, j)$, and each vertex is guaranteed to give rise to *exactly* one such pair. Letting $\mu_0^{(i,j)}(\mathcal{D}_0)$ refer to the multiplicity of a tuple in $\mathcal{D}_0$, we know that, since the label count is different, there is *at least* one tuple $(k, l)$ with $\mu_0^{(k,l)}(\mathcal{D}_0) \neq \mu_0^{(k,l)}(\mathcal{D}_0')$. Hence, $\mathcal{D}_0 \neq \mathcal{D}_0'$. $\qquad\square$

While Theorem 3 proves the *existence* of a filtration, the proof is constructive and relies on the existence of the WL[1] labels. Moreover, the resulting filtration is typically not *injective*, thus precluding the applicability of Theorem 1. The following Lemma discusses a potential fix for filtration functions of arbitrary dimensionality; our context is a special case of this.

**Lemma 1.** *For all $\epsilon > 0$ and $f: V \to \mathbb{R}^d$ there is an injective function $\tilde{f}$ such that $\|f - \tilde{f}\|_\infty < \epsilon$.*

*Proof.* Let $V = \{v_1, \ldots, v_n\}$ the vertices of a graph and $\operatorname{im} f = \{u_1, \ldots, u_m\} \subset \mathbb{R}^d$ their images under $f$. Since $f$ is not injective, we have $m < n$. We resolve non-injective vertex pairs iteratively. For $u \in \operatorname{im} f$, let $V' := \{v \in V \mid f(v) = u\}$. If $V'$ only contains a single element, we do not have to do anything. Otherwise, for each $v' \in V'$, pick a new value from $\mathrm{B}_\epsilon(u) \setminus \operatorname{im} f$, where $\mathrm{B}_r(x) \subset \mathbb{R}^d$ refers to the open ball of radius $r$ around a point $x$ (for our case, i.e. $d = 1$, this becomes an open interval in $\mathbb{R}$). Since we only ever remove a finite number of points, such a new value always exists, and we can modify $\operatorname{im} f$ accordingly. The number of vertex pairs for which $f$ is non-injective decreases by at least one in every iteration, hence after a finite number of iterations, we have modified $f$ to obtain $\tilde{f}$, an *injective* approximation to $f$. By always picking new values from balls of radius $\epsilon$, we ensure that $\|f - \tilde{f}\|_\infty < \epsilon$, as required. $\qquad\square$

Using this Lemma, we can ensure that the function $f$ in Theorem 3 is injective, thus ensuring differentiability according to Theorem 1. However, we need to make sure that the main message of Theorem 2 still holds, i.e. the two graphs $G$ and $G'$ must lead to different persistence diagrams even under the injective filtration function $\tilde{f}$. This is addressed by the following lemma, which essentially states that moving from $f$ to an injective $\tilde{f}$ does not result in coinciding persistence diagrams.

**Lemma 2.** *Let $G$ and $G'$ be two graphs whose $0$-dimensional persistence diagrams $\mathcal{D}_0$ and $\mathcal{D}_0'$ are calculated using a filtration function $f$ as described in Theorem 3. Moreover, given $\epsilon > 0$, let $\tilde{f}$ be an injective filtration function with $\|f - \tilde{f}\|_\infty$ and corresponding $0$-dimensional persistence diagrams $\widetilde{\mathcal{D}_0}$ and $\widetilde{\mathcal{D}_0'}$. If $\mathcal{D}_0 \neq \mathcal{D}_0'$, we also have $\widetilde{\mathcal{D}_0} \neq \widetilde{\mathcal{D}_0'}$.*

*Proof.* Since $\tilde{f}$ is injective, each tuple in $\widetilde{\mathcal{D}_0}$ and $\widetilde{\mathcal{D}'_0}$ has multiplicity $1$. But under $f$, there were differences in multiplicity for at least one tuple $(k, l)$. Hence, given $\tilde{f}$, there exists at least one tuple $(k, l) \in \widetilde{\mathcal{D}_0} \cup \widetilde{\mathcal{D}'_0}$ with $(k, l) \notin \widetilde{\mathcal{D}_0} \cap \widetilde{\mathcal{D}'_0}$. As a consequence, $\widetilde{\mathcal{D}_0} \neq \widetilde{\mathcal{D}'_0}$. □

For readers familiar with TDA, this lemma can also be proved in a simpler way by noting that if the bottleneck distance between $\mathcal{D}_0$ and $\mathcal{D}'_0$ is non-zero, it will remain so if one picks a perturbation that is sufficiently small (this fact can also be proved more rigorously). In any case, the preceding proofs enable us to finally state a stronger variant of the expressivity theorem, initially stated as Theorem 2 in the main paper. For consistency reasons with the nomenclature in this section, we slightly renamed the filtration function.

**Theorem 4** (Expressivity, strong version). *Persistent homology is* at least *as expressive as WL[1], i.e. if the WL[1] label sequences for two graphs $G$ and $G'$ diverge, there exists an injective filtration $\tilde{f}$ such that the corresponding $0$-dimensional persistence diagrams $\widetilde{\mathcal{D}_0}$ and $\widetilde{\mathcal{D}'_0}$ are not equal.*

*Proof.* We obtain a non-injective filtration function $f$ from Theorem 3, under which $\mathcal{D}_0$ and $\mathcal{D}'_0$ are not equal. By Lemma 1, for every $\epsilon > 0$, we can find an injective function $\tilde{f}$ with $\|f - \tilde{f}\|_\infty < \epsilon$. According to Lemma 2, the persistence diagrams calculated by this function do not coincide, i.e. $\widetilde{\mathcal{D}_0} \neq \widetilde{\mathcal{D}'_0}$. Hence, $\tilde{f}$ is a filtration function and, according to Theorem 1, differentiability is ensured. □

## C  Empirical Expressivity: Analysis of (Strongly) Regular Graphs

A *k-regular graph* is a graph $G = (V, E)$ in which all vertices have degree $k$. For $k = \{3, 4\}$, such graphs are also known as cubic and quartic graphs, respectively. The Weisfeiler–Lehman test is capable of distinguishing between certain variants of these graphs (even though we observe that WL[1] is not sufficient to do so). Similarly, a *strongly regular* graph is a graph $G = (V, E)$ with two integers $\lambda, \mu \in \mathbb{N}$ such that each pair of adjacent vertices has $\lambda$ common neighbours, whereas every pair of non-adjacent vertices has $\mu$ common neighbours.

Persistent homology can make use of higher-order connectivity information to distinguish between these data sets. To demonstrate this, we use a standard degree filtration and compute persistent homology of the graph, including all higher-order cliques. We then calculate the *total persistence* of each persistence diagram $\mathcal{D}$, and use it to assign a feature vector to the graph. This is in some sense the simplest way of employing persistent homology; notice that we are *not* learning a new filtration but keep a fixed one. Even in this scenario, we find that there is always a significant number of pairs of graphs whose feature vectors do not coincide—or, conversely speaking, as Table S4 shows, there are only between 14%–22% of pairs of graphs that we cannot distinguish by this simple scheme. This illustrates the general expressivity that a topology-based perspective can yield. For the strongly-regular graphs, we observe even lower error rates: we only fail to distinguish about 1.2% of all pairs (specifically, 8236 out of 7424731 pairs) of the 3854 strongly-regular graphs on 35 vertices with $\lambda = \mu = 9$ (McKay and Spence, 2001).

**A note on computational complexity.**   At the same time, this approach is also not without its disadvantages. Since we perform a clique counting operation, the complexity of the calculation increases considerably, and we would not suggest to use persistent homology of arbitrary dimensions in practice. While there are some algorithmic improvements in topological constructions (Zomorodian, 2010), naïve persistent homology calculations of arbitrary order may quickly become infeasible for larger data sets. This can be alleviated to a certain extent by *approximating* persistent homology (Cavanna et al., 2015a;b; Sheehy, 2013), but the practical benefits of this are unclear. Nevertheless, this experiment should therefore be considered as an indication of the utility of topological features in general to complement and enhance existing architectures.

## D  Experimental Setup & Computational Resources

Following the setup of Dwivedi et al. (2020), we implemented the following training procedure: All models are initialised with an initial learning rate $lr_{init}$, which varies between the data sets. During

Table S4: Error rates when using persistent homology with a degree filtration to classify pairs of $k$-regular on $n$ vertices. R3-N12 denotes 3-regular graphs on 12 vertices, for instance. This list is by no means exhaustive, but indicates the general utility of persistent homology and its filtration-based analysis.

| Data set | Graphs | Pairs | Error | Error rate |
|---|---|---|---|---|
| R3-N12 | 85 | 3570 | 712 | 19.94% |
| R3-N14 | 509 | 129286 | 26745 | 20.69% |
| R3-N16 | 4060 | 8239770 | 1757385 | 21.33% |
| R4-N10 | 59 | 1711 | 229 | 13.38% |
| R4-N11 | 265 | 34980 | 4832 | 13.81% |
| R4-N12 | 1544 | 1191196 | 170814 | 14.34% |

Table S5: Parameters of learning rate scheduling (including 'patience' parameters) for training of models in this work.

| NAME | MNIST, CIFAR-10, PATTERNS, CLUSTER | PROTEINS, ENZYMES, DD |
|---|---|---|
| $lr_{init}$ | $1 \times 10^{-3}$ | $7 \times 10^{-4}$ |
| $lr_{min}$ | $1 \times 10^{-5}$ | $1 \times 10^{-6}$ |
| $lr_{patience}$ | 10 | 25 |

training the loss on the validation split is monitored and the learning rate is halved if the validation loss does not improve over a period of $lr_{patience}$. Runs are stopped when the learning rate gets reduced to a value of lower than $lr_{min}$. The parameters for the different data sets are shown in Table S5.

Our method is implemented in `Python`, making heavy use of the `pytorch-geometric` library (Fey and Lenssen, 2019), licensed under the MIT License, and the `pytorch-lightning` library (Falcon et al., 2019), licensed under the Apache 2.0 License. The training and hyperparameter selection was performed using 'Weights and Biases' (Biewald, 2020), resulting in additional reports, tables, and log information, which will simplify the reproducibility of this work. We will make our own code available, using either the MIT License or a BSD 3-Clause license, which precludes endorsing/promoting our work without prior approval. All licenses used for the code are compatible with this licensing choice.

As for the data sets, we use data sets that are available in `pytorch-geometric` for graph learning tasks. Some of the benchmark data sets have been originally provided by Morris et al. (2020), others (CIFAR-10, CLUSTER, MNIST, PATTERN) have been provided by Dwivedi et al. (2020) in the context of a large-scale graph neural network benchmarking effort.

**Ablated static variant of TOGL.** Next to all the experiments presented in the main paper, we also developed a *static variant* of our layer, serving as an additional ablation method (this nomenclature will be used in all supplementary tables). The static variant mimics the calculation of a filtration in terms of the number of parameters but without taking topological features into account. The layer uses a *static* mapping (instead of a dynamic one based on persistent homology) of vertices to themselves (for dimension 0), and employs a random edge selection process (for dimension 1). This has the effect of learning graph-level information that is not strictly based on topology but on node feature values. The static variant of TOGL reduces to an MLP that is applied per vertex in case interactions between tuples are not considered, and to the application of a `DeepSet` Zaheer et al. (2017) to the vertex representations if interactions between tuples are incorporated. Generally, if the static variant performs well on a data set, we assume that performance is driven much more by the availability of *any* graph-level type of information, such as the existence of certain nodes or groups of nodes, as opposed to topological information.

**Compute resources.** Most of the jobs were run on our internal cluster, comprising 64 physical cores (`Intel(R) Xeon(R) CPU E5-2620 v4 @ 2.10GHz`) with 8 GeForce GTX 1080 GPUs. A smaller fraction has been run on another cluster, containing 40 physical cores (`Intel(R)`

Table S6: The set of hyperparameters that we use to train TOGL, along with their respective value ranges. Notice that 'dropout' could be made configurable, but this would make our setup incomparable to the setup proposed by Dwivedi et al. (2020) for benchmarking GNNs.

| NAME | VALUE(S) |
|---|---|
| `DeepSet` | {True, False} |
| Depth | $\{3, 4\}$ |
| Dim1 | True (by default, we always use cycle information) |
| Dropout | 0. |
| Early Topo | (True, False) |
| Static | {True, False} (to evaluate the static variant) |
| Filtration Hidden Dimension | 32 |
| Hidden Dimension | 138–146 |
| No. coordinate functions | 3 |
| No. filtrations | 8 |
| Residual and Batch Norm | True |
| Share filtration parameters | True |

`Xeon(R) CPU E5-2630L v4 @ 1.80GHz`) with 2 Quadro GV100 GPUs and 1 Titan XP GPU.

## E HYPERPARAMETERS

Table S6 contains a listing of all hyperparameters used to train TOGL. For the Weisfeiler–Lehman subtree kernel, which we employed as a comparison partner on the synthetic data sets, we used a support vector machine classifier with a linear kernel, whose regularisation parameter $C \in \{10^{-4}, 10^{-3}, \ldots, 10^4\}$ is trained via 5-fold cross validation, which is repeated 10 times to obtain standard deviations. This follows closely the setup in graph classification literature.

## F EXTENDED RESULTS FOR THE SYNTHETIC DATA SETS

In the main paper, we depicted a concise analysis of synthetic data sets (Figure 1). Here, we provide a more detailed ablation of this data set, highlighting the differences between TOGL and its static baseline, but also disentangling the performance of 0-dimensional and 1-dimensional topological features, i.e. connected components and cycles, respectively. Figure S4 depicts the results. As stated in the main paper, we observe that (i) both cycles and connected components are crucial for reaching high predictive performance, and (ii) the static variant is performing slightly better than the standard GCN, but *significantly worse* than TOGL, due to its very limited access to topological information.

## G VISUALISATION OF THE LEARNT FILTRATIONS

TOGL can learn an arbitrary number of filtrations on the graphs. In Figure S7, we display 3 different filtrations learnt on an randomly picked graph from the DD data set and compare it against the classical node degree filtration. The width of the nodes is plotted proportional to the node degree filtration while the colour saturation is proportional to the learnt filtration. Filtration 0 appears to be correlated with the degree filtration, while other filtrations learnt by TOGL seem to focus on different local properties of the graph.

## H EXPERIMENTS ON LAYER PLACEMENT

A priori, it is not clear where to place TOGL. We therefore investigated the impact of putting TOGL *first* (thus imbuing the graph with topological features arising from the neighbours of nodes) or at another position in the network. In Table Table S7, we investigate the performance of the layer when TOGL is placed at different positions in a GNN architecture composed of 3 graph convolution layers. It appears that for this dataset, positioning TOGL before the last GNN layer leads to best performance

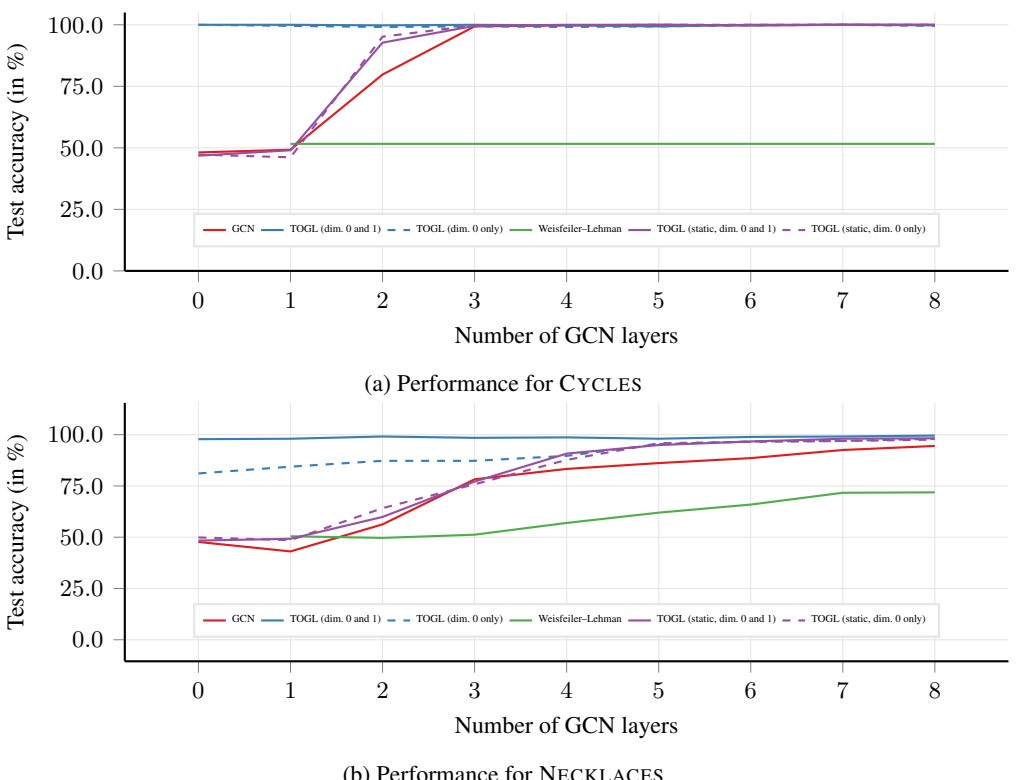

(a) Performance for CYCLES

(b) Performance for NECKLACES

Figure S4: Performance comparison on synthetic data sets as a function of the number of GCN layers or Weisfeiler–Lehman iterations. This is an extended version of Figure 1. For TOGL, we show the performance with respect to the dimensionality of topological features that are being used. Since the standard deviations are negligible, we refrain from showing them here.

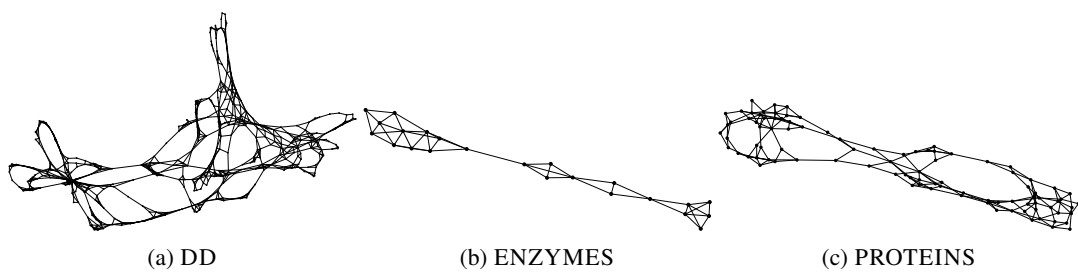

(a) DD            (b) ENZYMES            (c) PROTEINS

Figure S5: Example graphs of the benchmark data sets that describe molecular structures. These graphs give rise to complex topological structures that can be exploited.

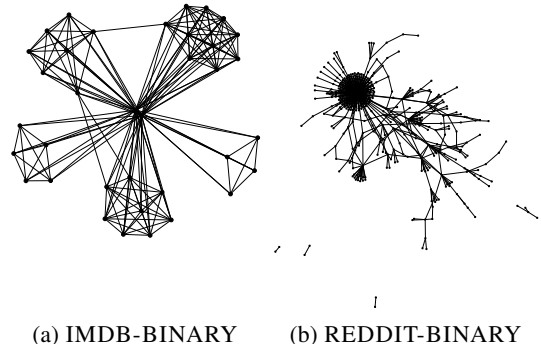

(a) IMDB-BINARY      (b) REDDIT-BINARY

Figure S6: Example graphs of the benchmark data sets that correspond to extracted social networks. These graphs are best described in terms of clique connectivity; the existence of a single 'hub' node does not give rise to a complex topological structures that can be exploited *a priori*.

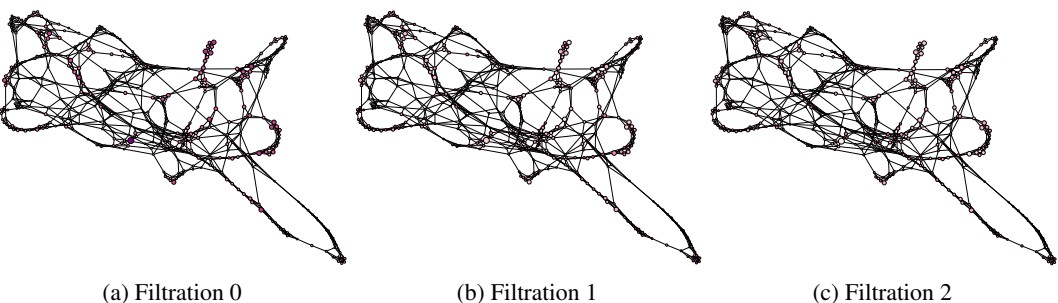

(a) Filtration 0      (b) Filtration 1      (c) Filtration 2

Figure S7: Examples of different filtrations jointly learnt on an example graph randomly picked from the DD data set. The width of each node dot is proportional to its node degree, the colour saturation is proportional to the filtration value.

Table S7: Test accuracies for different layer positions. TOGL can be placed at different positions. When placed at position $0$, all downstream layers incorporate topological information, but only as much as can be gleaned from the nodes of graph. By contrast, placing TOGL *last* makes a `readout` function that incorporates information from multiple filtrations.

| LAYER POSITION | METHOD | NECKLACES |
|---|---|---|
| 3 | GCN-3-TOGL-1 (no interaction) | $97.17 \pm 0.4$ |
|   | GCN-3-TOGL-1 (static - no interaction) | $94.5 \pm 2.9$ |
| 2 | GCN-3-TOGL-1 (no interaction) | $98.8 \pm 0.8$ |
|   | GCN-3-TOGL-1 (static - no interaction) | $77.2 \pm 4.5$ |
| 1 | GCN-3-TOGL-1 (no interaction) | $97.4 \pm 1.8$ |
|   | GCN-3-TOGL-1 (static - no interaction) | $66.3 \pm 5.8$ |
| 0 | GCN-3-TOGL-1 (no interaction) | $98.6 \pm 2.1$ |
|   | GCN-3-TOGL-1 (static - no interaction) | $61.8 \pm 5.3$ |

Table S8: Test accuracies for different layer positions. TOGL can be placed at different positions. When placed at position $0$, all downstream layers incorporate topological information, but only as much as can be gleaned from the nodes of graph. By contrast, placing TOGL *last* makes a `readout` function that incorporates information from multiple filtrations.

| LAYER POSITION | METHOD | IMDB-BINARY | REDDIT-BINARY | PROTEINS-FULL | DD |
|---|---|---|---|---|---|
| 4 | GCN-3-TOGL-1 (interaction) | $74.9 \pm 4.0$ | $92.7 \pm 1.4$ | $74.9 \pm 3.3$ | $71.5 \pm 4.5$ |
|   | GCN-3-TOGL-1 (no interaction) | $71.6 \pm 2.1$ | $89.4 \pm 2.1$ | $75.9 \pm 4.0$ | $74.8 \pm 2.0$ |
|   | GCN-3-TOGL-1 | $73.4 \pm 3.2$ | $90.0 \pm 2.8$ | $75.6 \pm 4.0$ | $72.3 \pm 4.6$ |
|   | GCN-3-TOGL-1 (static - interaction) | $74.2 \pm 3.7$ | $91.9 \pm 1.6$ | $75.2 \pm 2.7$ | $71.1 \pm 5.1$ |
|   | GCN-3-TOGL-1 (static - no interaction) | $73.8 \pm 4.8$ | $89.4 \pm 2.1$ | $75.8 \pm 4.0$ | $70.9 \pm 2.6$ |
|   | GCN-3-TOGL-1 (static) | $74.2 \pm 4.7$ | $92.3 \pm 2.3$ | $75.2 \pm 3.9$ | $70.3 \pm 5.0$ |
| 3 | GCN-3-TOGL-1 (interaction) | $70.6 \pm 5.6$ | $92.2 \pm 1.3$ | $75.5 \pm 3.1$ | $73.0 \pm 2.8$ |
|   | GCN-3-TOGL-1 (no interaction) | $76.1 \pm 3.9$ | $90.4 \pm 1.6$ | $75.7 \pm 3.1$ | $75.5 \pm 3.4$ |
|   | GCN-3-TOGL-1 | $74.8 \pm 5.7$ | $91.0 \pm 1.7$ | $73.9 \pm 3.4$ | $73.9 \pm 3.4$ |
|   | GCN-3-TOGL-1 (static - interaction) | $72.0 \pm 3.0$ | $92.9 \pm 1.2$ | $76.0 \pm 2.1$ | $71.5 \pm 5.8$ |
|   | GCN-3-TOGL-1 (static - no interaction) | $74.0 \pm 6.8$ | $90.8 \pm 4.4$ | $75.5 \pm 3.8$ | $71.8 \pm 4.1$ |
|   | GCN-3-TOGL-1 (static) | $73.4 \pm 6.2$ | $92.2 \pm 1.3$ | $75.8 \pm 2.7$ | $71.6 \pm 4.5$ |
| 2 | GCN-3-TOGL-1 (interaction) | $73.2 \pm 2.1$ | $91.7 \pm 0.5$ | $75.4 \pm 2.9$ | $74.2 \pm 2.7$ |
|   | GCN-3-TOGL-1 (no interaction) | $74.9 \pm 2.3$ | $87.6 \pm 4.1$ | $75.5 \pm 4.3$ | $74.7 \pm 4.3$ |
|   | GCN-3-TOGL-1 | $74.8 \pm 1.9$ | $89.6 \pm 2.2$ | $75.4 \pm 4.1$ | $74.9 \pm 3.5$ |
|   | GCN-3-TOGL-1 (static - interaction) | $72.2 \pm 4.4$ | $91.6 \pm 2.9$ | $76.0 \pm 4.1$ | $72.7 \pm 3.9$ |
|   | GCN-3-TOGL-1 (static - no interaction) | $72.4 \pm 4.3$ | $87.6 \pm 4.1$ | $74.4 \pm 4.1$ | $70.7 \pm 3.4$ |
|   | GCN-3-TOGL-1 (static) | $73.6 \pm 4.3$ | $92.4 \pm 0.8$ | $74.5 \pm 3.8$ | $71.5 \pm 4.1$ |
| 1 | GCN-3-TOGL-1 (interaction) | $69.6 \pm 4.3$ | $91.1 \pm 0.7$ | $75.7 \pm 2.6$ | $74.1 \pm 4.1$ |
|   | GCN-3-TOGL-1 (no interaction) | $71.8 \pm 2.7$ | $89.6 \pm 1.7$ | $75.3 \pm 4.2$ | $75.6 \pm 3.5$ |
|   | GCN-3-TOGL-1 | $70.6 \pm 3.5$ | $89.9 \pm 1.6$ | $75.2 \pm 3.9$ | $74.7 \pm 3.0$ |
|   | GCN-3-TOGL-1 (static - interaction) | $73.4 \pm 4.6$ | $90.5 \pm 1.4$ | $76.3 \pm 3.4$ | $73.2 \pm 3.6$ |
|   | GCN-3-TOGL-1 (static - no interaction) | $72.4 \pm 3.1$ | $89.6 \pm 1.7$ | $75.2 \pm 3.4$ | $72.7 \pm 3.8$ |
|   | GCN-3-TOGL-1 (static) | $73.4 \pm 4.6$ | $90.1 \pm 0.9$ | $75.6 \pm 3.8$ | $72.5 \pm 4.3$ |

when the non-static version is considered. Importantly, this contrasts with the approach of Hofer et al. (2019), where topological information is only available at the read-out level, which would lead to the worst performance on this dataset.

We complemented our experiments on layer placement with an exhaustive assessment of the performance of our model on on the IMDB-Binary, Reddit-Binary, Proteins and DD datasets. Please refer to Table S8 for these results.

# I EXTENDED EXPERIMENTS FOR STRUCTURED-BASED GRAPH CLASSIFICATION

This section contains more results for the structured-based classification of graph data sets shown in Section 5.3 and Figure 3. Table Table S9 contains a detailed listing of performance values under

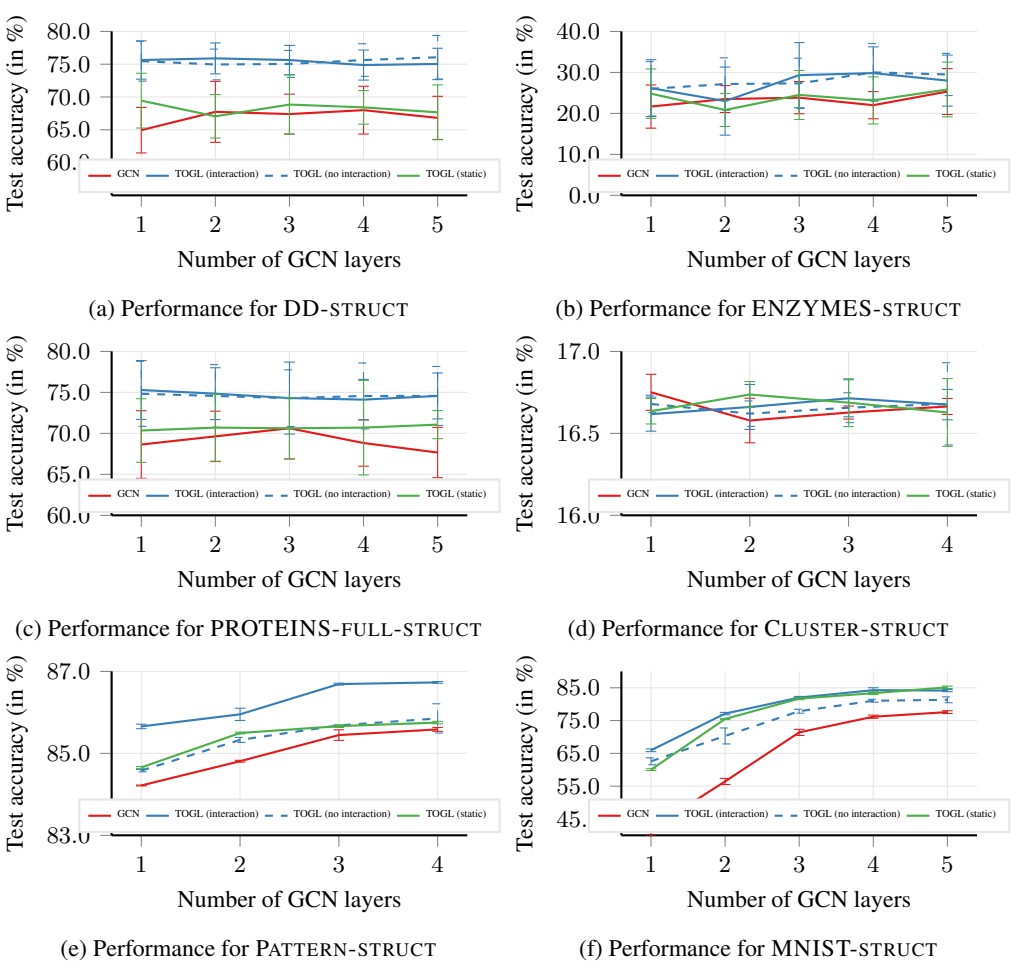

(a) Performance for DD-STRUCT

(b) Performance for ENZYMES-STRUCT

(c) Performance for PROTEINS-FULL-STRUCT

(d) Performance for CLUSTER-STRUCT

(e) Performance for PATTERN-STRUCT

(f) Performance for MNIST-STRUCT

Figure S8: Comparison of test accuracy for the structure-based variants of the benchmark data sets while varying network depth, i.e. the number of GCN layers. Error bars denote the standard deviation of test accuracy over 10 cross-validation folds. *Interaction* refers to using a `DeepSets` approach for embedding persistence diagrams, whereas *no interaction* uses persistence diagrams coordinate functions that do not account for pairwise interactions.

certain additional ablation scenarios, such as a disentangled comparison of the performance with respect to picking a certain type of embedding scheme for topological features, i.e. deep sets versus known embedding functions for persistence diagrams, the latter of which cannot handle *interactions* between tuples in a diagram. The version labelled *General* consist of non-static versions where the type of feature embedding scheme is considered as an hyper-parameter. We then report the test accuracy of the version whose validation accuracy was highest. The *General -Static* variant is similar but for the *static* version of TOGL. Figure S8 depicts the performance on different data sets, focusing on a comparison between the different embedding types and the static variant of our layer.

The behaviour of our static variant indicates that predictive performance is *not* driven by having access to topological information on the benchmark data sets. To some extent, this is surprising, as some of the data sets turn out to contain salient topological features even visual inspection (Figure S5), whereas the social networks exhibit more community-like behaviour (Figure S6). Together with the results from Section 5.3, this demonstrates that existing data sets often 'leak' structural information through their labels/features. Yet, for the node classification data sets CLUSTER and PATTERN, GCN-3-TOGL-1 or its static variant perform best among all comparison partners.[4]

---

[4]On these data sets, our results on GCN-4 differ slightly due to a known misalignment in the implementation of Dwivedi et al. (2020), as compared to the original GCN architecture.

Table S9: Test Accuracies for different structural data sets.

| Data Set | Depth | GCN | GIN | GAT | GCN-TOGL (General) | GCN-TOGL (General - Static) | GCN-TOGL (No Interaction) | GCN-TOGL (Interaction) | GCN-TOGL (Static - No Interaction) | GCN-TOGL (Static-Interaction) |
|---|---|---|---|---|---|---|---|---|---|---|
| DD | 1 | 64.9 ± 3.5 | 75.0 ± 2.3 | 59.8 ± 2.0 | **76.1 ± 2.7** | 68.8 ± 4.1 | 75.5 ± 3.0 | 75.6 ± 2.9 | 69.4 ± 4.2 | 66.7 ± 3.2 |
| DD | 2 | 67.7 ± 4.7 | 75.4 ± 3.1 | 61.6 ± 4.3 | 74.8 ± 2.7 | 66.5 ± 2.8 | 75.0 ± 2.3 | **75.9 ± 2.4** | 67.1 ± 3.3 | 66.5 ± 3.2 |
| DD | 3 | 67.4 ± 3.0 | 74.9 ± 3.3 | 62.0 ± 4.3 | 75.3 ± 2.4 | 68.8 ± 4.5 | 75.0 ± 2.1 | **75.6 ± 2.2** | 68.8 ± 4.5 | 65.6 ± 2.2 |
| DD | 4 | 68.0 ± 3.6 | **75.6 ± 2.8** | 63.3 ± 3.7 | 75.1 ± 2.1 | 68.0 ± 2.4 | 75.6 ± 2.5 | 74.9 ± 2.3 | 68.4 ± 2.6 | 64.4 ± 4.8 |
| DD | 5 | 66.8 ± 3.3 | 75.6 ± 2.1 | 62.6 ± 3.9 | 75.9 ± 3.3 | 67.6 ± 4.1 | **76.1 ± 3.3** | 75.0 ± 2.4 | 67.7 ± 4.2 | 68.3 ± 3.6 |
| Proteins | 1 | 68.6 ± 4.1 | 72.0 ± 3.4 | 66.8 ± 3.5 | **75.3 ± 3.4** | 70.3 ± 2.6 | 74.8 ± 4.0 | 75.3 ± 3.6 | 70.3 ± 3.9 | 71.0 ± 3.6 |
| Proteins | 2 | 69.6 ± 3.1 | 73.1 ± 4.1 | 66.8 ± 3.2 | **74.9 ± 3.5** | 71.0 ± 3.5 | 74.6 ± 3.8 | 74.8 ± 3.2 | 70.7 ± 4.1 | 71.2 ± 4.7 |
| Proteins | 3 | 70.6 ± 3.7 | 73.0 ± 4.1 | 67.0 ± 3.4 | 73.8 ± 4.3 | 70.5 ± 3.7 | 74.3 ± 3.5 | **74.3 ± 4.4** | 70.6 ± 3.7 | 70.0 ± 3.1 |
| Proteins | 4 | 68.8 ± 2.8 | 74.6 ± 3.1 | 67.5 ± 2.6 | 73.8 ± 3.7 | 71.2 ± 5.1 | **74.6 ± 4.0** | 74.1 ± 2.5 | 70.7 ± 5.8 | 70.6 ± 3.1 |
| Proteins | 5 | 67.7 ± 3.1 | 73.0 ± 3.2 | 65.7 ± 1.7 | **74.7 ± 3.3** | 71.2 ± 1.9 | 74.6 ± 3.6 | 74.6 ± 2.8 | 71.1 ± 1.7 | 71.1 ± 3.7 |
| Enzymes | 1 | 21.7 ± 5.3 | 20.8 ± 4.2 | 15.3 ± 3.1 | **26.3 ± 6.4** | 23.0 ± 6.3 | 26.0 ± 6.6 | 26.2 ± 6.9 | 24.8 ± 6.0 | 21.8 ± 4.0 |
| Enzymes | 2 | 23.5 ± 3.3 | 20.5 ± 5.3 | 17.5 ± 5.2 | 26.2 ± 8.1 | 21.2 ± 4.6 | **27.2 ± 6.4** | 23.0 ± 8.3 | 20.8 ± 4.0 | 21.5 ± 5.9 |
| Enzymes | 3 | 23.8 ± 3.9 | 20.5 ± 6.0 | 16.3 ± 7.0 | 29.2 ± 6.1 | 24.3 ± 6.1 | 27.3 ± 6.1 | **29.3 ± 7.9** | 24.5 ± 6.0 | 21.2 ± 3.4 |
| Enzymes | 4 | 22.0 ± 3.3 | 21.3 ± 6.5 | 21.7 ± 2.9 | **30.3 ± 6.5** | 23.7 ± 5.4 | 30.0 ± 7.0 | 29.8 ± 6.4 | 23.2 ± 5.7 | 22.7 ± 4.2 |
| Enzymes | 5 | 25.3 ± 5.6 | 21.0 ± 4.4 | 19.8 ± 5.8 | 29.0 ± 5.2 | 26.8 ± 7.2 | **29.5 ± 5.2** | 28.0 ± 6.2 | 25.8 ± 6.7 | 26.8 ± 5.3 |
| Pattern | 1 | 84.2 ± 0.0 | 69.5 ± 0.0 | 55.2 ± 3.6 | **85.7 ± 0.0** | 84.6 ± 0.0 | 84.6 ± 0.0 | 85.7 ± 0.1 | 84.7 ± 0.0 | 84.2 ± 0.0 |
| Pattern | 2 | 84.8 ± 0.0 | 84.3 ± 0.0 | 55.1 ± 6.3 | **86.1 ± 0.0** | 85.5 ± 0.0 | 85.3 ± 0.1 | 86.0 ± 0.1 | 85.5 ± 0.0 | 84.9 ± 0.0 |
| Pattern | 3 | 85.4 ± 0.1 | 84.7 ± 0.0 | 62.9 ± 5.2 | **86.7 ± 0.0** | 85.7 ± 0.0 | 85.7 ± 0.0 | 86.7 ± 0.0 | 85.7 ± 0.0 | 85.6 ± 0.0 |
| Pattern | 4 | 85.6 ± 0.0 | 84.8 ± 0.0 | 58.3 ± 8.8 | **86.7 ± 0.0** | 85.8 ± 0.0 | 85.9 ± 0.4 | 86.7 ± 0.0 | 85.8 ± 0.0 | 85.7 ± 0.0 |
| Cluster | 1 | **16.8 ± 0.1** | 16.7 ± 0.1 | 16.7 ± 0.1 | 16.6 ± 0.0 | 16.7 ± 0.0 | 16.7 ± 0.1 | 16.6 ± 0.1 | **16.7 ± 0.1** | 16.6 ± 0.1 |
| Cluster | 2 | 16.6 ± 0.1 | 16.6 ± 0.1 | 16.6 ± 0.1 | 16.6 ± 0.0 | 16.6 ± 0.0 | 16.6 ± 0.1 | 16.7 ± 0.1 | **16.7 ± 0.1** | 16.6 ± 0.1 |
| Cluster | 3 | 16.6 ± 0.0 | 16.4 ± 0.2 | 16.7 ± 0.1 | 16.6 ± 0.0 | 16.6 ± 0.0 | 16.7 ± 0.1 | **16.7 ± 0.1** | 16.7 ± 0.1 | 16.6 ± 0.2 |
| Cluster | 4 | 16.7 ± 0.0 | 16.4 ± 0.1 | 16.7 ± 0.0 | 16.8 ± 0.0 | **16.8 ± 0.0** | 16.7 ± 0.3 | 16.7 ± 0.1 | 16.6 ± 0.2 | 16.6 ± 0.3 |
| MNIST | 1 | 42.0 ± 2.2 | 42.9 ± 0.0 | 40.4 ± 2.9 | **66.4 ± 0.0** | 60.3 ± 0.0 | 62.6 ± 1.1 | 65.9 ± 0.4 | 60.0 ± 0.3 | 56.4 ± 0.4 |
| MNIST | 2 | 56.4 ± 0.9 | 68.3 ± 0.7 | 48.1 ± 7.3 | **77.4 ± 0.0** | 75.3 ± 0.0 | 70.3 ± 2.4 | 77.1 ± 0.4 | 75.5 ± 0.2 | 70.5 ± 0.9 |
| MNIST | 3 | 71.4 ± 0.9 | 79.1 ± 0.3 | 68.7 ± 2.6 | **82.3 ± 0.0** | 81.3 ± 0.0 | 77.9 ± 0.6 | 82.0 ± 0.3 | 81.7 ± 0.3 | 78.8 ± 0.8 |
| MNIST | 4 | 76.2 ± 0.5 | 83.4 ± 0.9 | 63.2 ± 10.4 | **84.8 ± 0.0** | 82.9 ± 0.0 | 81.0 ± 0.4 | 84.3 ± 0.7 | 83.4 ± 0.4 | 82.0 ± 0.9 |
| MNIST | 5 | 77.6 ± 0.4 | 85.1 ± 0.6 | 56.8 ± 20.4 | 84.0 ± 0.0 | 85.0 ± 0.0 | 81.3 ± 0.9 | 84.2 ± 0.3 | **85.1 ± 0.4** | 83.1 ± 0.5 |

Table S10: Results for the structure-based experiments. We depict the test accuracy obtained on various benchmark data sets when only considering structural information (i.e. the network has access to *uninformative* node features). Graph classification results are shown on the left, while node classification results are shown on the right.

| | *Graph classification* | | | | *Node classification* | |
|---|---|---|---|---|---|---|
| METHOD | DD | ENZYMES | MNIST | PROTEINS | CLUSTER | PATTERN |
| GAT-4 | 63.3±3.7 | 21.7± 2.9 | 63.2±10.4 | 67.5± 2.6 | 16.7± 0.0 | 58.3±8.8 |
| GIN-4 | 75.6±2.8 | 21.3± 6.5 | 83.4± 0.9 | **74.6**± 3.1 | 16.4± 0.1 | 84.8±0.0 |
| GCN-4 (*baseline*) | 68.0±3.6 | 22.0± 3.3 | 76.2± 0.5 | 68.8± 2.8 | 16.7± 0.0 | 85.6±0.0 |
| GCN-3-TOGL-1 | 75.1±2.1 | **30.3± 6.5** | **84.8± 0.4** | 73.8± 4.3 | **16.8± 0.0** | **86.7±0.0** |
| GCN-3-TOGL-1 (static) | 68.0±2.4 | 23.7± 5.4 | 82.9± 0.0 | 71.2± 5.1 | **16.8± 0.0** | 85.8±0.0 |
| GIN-3-TOGL-1 | 76.2±2.4 | 23.7± 6.9 | 84.4± 1.1 | 73.9± 4.9 | 16.6± 0.3 | 86.6±0.1 |
| GIN-3-TOGL-1 (static) | **76.3±2.8** | 25.2± 7.0 | 83.9± 0.1 | 74.2± 4.2 | 16.4± 0.1 | 85.4±0.1 |
| GAT-3-TOGL-1 | 75.7±2.1 | 23.5± 6.1 | 77.2±10.5 | 72.4± 4.6 | 16.5± 0.1 | 75.9±3.1 |
| GAT-3-TOGL-1 (static) | 68.4±3.4 | 22.7± 3.9 | 81.9± 1.1 | 68.9± 4.0 | 16.7± 0.0 | 60.5±3.0 |

Table S11: Test accuracy on benchmark data sets (following standard practice, we report weighted accuracy on CLUSTER and PATTERN). Methods printed in black have been run in our setup, while methods printed in grey are cited from the literature, i.e. Dwivedi et al. (2020), Morris et al. (2020) for IMDB-B and REDDIT-B, and Borgwardt et al. (2020) for WL/WL-OA results GIN-4 results printed in *italics* are 1-layer GIN-$\epsilon$, as reported in Morris et al. (2020). Graph classification results are shown on the left, while node classification results are shown on the right.

| | *Graph classification* | | | | | | | *Node classification* | |
|---|---|---|---|---|---|---|---|---|---|
| METHOD | CIFAR-10 | DD | ENZYMES | MNIST | PROTEINS-FULL | IMDB-B | REDDIT-B | CLUSTER | PATTERN |
| GAT-4 | 64.2 ± 0.4 | **75.9 ± 3.8** | **68.5 ± 5.2** | 95.5 ± 0.2 | 76.3 ± 2.4 | — | — | 57.7 ± 0.3 | 75.8 ± 1.8 |
| GATED-GCN-4 | **67.3 ± 0.3** | 72.9 ± 2.1 | 65.7 ± 4.9 | **97.3 ± 0.1** | **76.4 ± 2.9** | — | — | 60.4 ± 0.4 | 84.5 ± 0.1 |
| GIN-4 | 55.5 ± 1.5 | 71.9 ± 3.9 | 65.3 ± 6.8 | 96.5 ± 0.3 | 74.1 ± 3.4 | *72.9 ± 4.7* | 89.8 ± 2.2 | 58.4 ± 0.2 | 85.6 ± 0.0 |
| WL | — | 77.7 ± 2.0 | 54.3 ± 0.9 | — | 73.1 ± 0.5 | 71.2 ± 0.5 | 78.0 ± 0.6 | — | — |
| WL-OA | — | 77.8 ± 1.2 | 58.9 ± 0.9 | — | 73.5 ± 0.9 | 74.0 ± 0.7 | 87.6 ± 0.3 | — | — |
| GCN-4 (*baseline*) | 54.2 ± 1.5 | 72.8 ± 4.1 | 65.8 ± 4.6 | 90.0 ± 0.3 | 76.1 ± 2.4 | 68.6 ± 4.9 | **92.8 ± 1.7** | 57.0 ± 0.9 | 85.5 ± 0.4 |
| GCN-3-TOGL-1 | 61.7 ± 1.0 | 73.2 ± 4.7 | 53.0 ± 9.2 | *95.5 ± 0.2* | 76.0 ± 3.9 | 72.0 ± 2.3 | 89.4 ± 2.2 | 60.4 ± 0.2 | 86.6 ± 0.1 |
| GCN-3-TOGL-1 (static) | 62.1 ± 0.5 | 71.0 ± 2.8 | 49.8 ± 7.0 | 94.8 ± 0.1 | 75.7 ± 3.6 | 72.8 ± 5.4 | 92.1 ± 1.6 | 60.5 ± 0.2 | 85.6 ± 0.1 |
| GIN-3-TOGL-1 | 61.3 ± 0.4 | 75.2 ± 4.2 | 43.8 ± 7.9 | 96.1 ± 0.1 | 73.6 ± 4.8 | **74.2 ± 4.2** | *89.7 ± 2.5* | 60.4 ± 0.2 | **86.7 ± 0.1** |
| GIN-3-TOGL-1 (static) | 61.8 ± 0.6 | 72.2 ± 5.3 | 43.3 ± 8.3 | 96.4 ± 0.1 | 74.7 ± 3.1 | 73.8 ± 2.4 | 89.1 ± 4.4 | **60.6 ± 0.3** | 85.5 ± 0.1 |
| GAT-3-TOGL-1 | 52.8 ± 3.4 | 73.7 ± 2.9 | 51.5 ± 7.3 | 0.0 ± 0.0 | 75.2 ± 3.9 | 70.8 ± 8.0 | 82.5 ± 8.7 | 58.4 ± 3.7 | 59.6 ± 3.3 |
| GAT-3-TOGL-1 (static) | 50.8 ± 3.6 | 72.8 ± 3.4 | 55.2 ± 9.1 | 96.2 ± 0.3 | 74.4 ± 2.5 | 68.7 ± 9.4 | 70.1 ± 9.9 | 58.7 ± 2.2 | 64.5 ± 14.2 |

Table S12: Test accuracy of the different methods on the Spheres vs. Torus classification task. We compare two architectures (GCN-4, GIN-4) with corresponding models where one layer is replaced with TOGL and highlight the winner of each comparison in **bold**.

| METHOD | SPHERES VS. TORUS |
|---|---|
| GCN-4 | 77.8±2.1 |
| GCN-3-TOGL-1 | **98.5±1.3** |
| GIN-4 | 76.2±3.5 |
| GIN-3-TOGL-1 | **99.6±0.9** |

Table S13: Graph classification results for the structure-based experiments. We depict the test accuracy obtained on various benchmark data sets when only considering structural information (i.e. the network has access to *uninformative* node features). We compare two architectures (GCN-4, GIN-4) with corresponding models where one layer is replaced with TOGL and highlight the winner of each comparison in **bold**.

| | *Graph classification* | | |
|---|---|---|---|
| METHOD | DD | PROTEINS | CIFAR-10 |
| GCN-6 | 70.5±4.9 | 70.1±4.0 | 45.4±0.5 |
| GCN-5-TOGL-1 | **75.9±2.5** | **73.5±3.5** | **54.4±0.7** |
| GCN-8 | 68.5±3.6 | 72.0±3.3 | 45.6±0.5 |
| GCN-7-TOGL-1 | **75.6±2.9** | **73.5±3.3** | **54.1±1.1** |
| GIN-6 | 76.1±2.4 | **72.9±3.4** | 53.6±1.4 |
| GIN-5-TOGL-1 | **76.5±2.0** | 72.1±3.6 | **54.1±1.2** |
| GIN-8 | **76.5±2.4** | **72.9±4.0** | 51.4±2.4 |
| GIN-7-TOGL-1 | 76.0±3.6 | 72.6±4.1 | **55.3±0.7** |

## J    GEOMETRIC DATASET

In this section, we showcase the potential of TOGL for improving performance on tasks of geometric data sets. Tasks such as 3D object recognition are gathering increasing attention in the machine learning community and our approach represents a promising direction to explore that area. We consider a simple synthetic data set consisting of graphs linking points on geometrical objects. In particular, we create a data set with two classes where one consists of graphs lying on a sphere and the other of graphs lying on a torus. The nodes of these graphs are composed of points sampled on the geometrical objects and embedded in an higher dimensional space. The graph is then constructed as a fully connected graphs between all sampled points. Table S12 presents the classification accuracy of different methods on this experiment. We observe that substituting a GNN layer with a TOGL layer allows to obtain nearly perfect accuracy, showcasing the crucial importance of a topology-informed layer in geometric data sets.

## K    LARGER NUMBER OF LAYERS IN THE GNN ARCHITECTURE

In this section, we investigate the performance of our approach. on architecture with a larger number of layers. For computational reasons, we limit our analysis to 3 data sets, namely DD, PROTEINS and CIFAR-10. We re-use the experimental setup as detailed in Section 5.1. Tables S13 and S14 show the classification accuracy for the structural and the benchmark setup respectively. What we observe is that the advantage of including TOGL is generally conserved, and similar to the one observed in the 4 layers case. In absolute terms, for the structural dataset, the performance on the DD dataset increases with more layers (for both the baseline and the TOGL version) but decreases for the PROTEINS dataset. On the benchmark dataset (including node features), the performance tends to increase for all three datasets compared to the 4-layers version.

Table S14: Test accuracy on benchmark data sets. Graph classification results are shown on the left; node classification results are shown on the right. Following Table 1, we take existing architectures and replace their second layer by TOGL; we use *italics* to denote the winner of each comparison. A **bold** value indicates the overall winner of a column, i.e. a data set.

| METHOD | *Graph classification* | | |
| | CIFAR-10 | DD | PROTEINS-FULL |
| --- | --- | --- | --- |
| GCN-6 | 53.6±0.6 | 72.7±2.8 | **76.3±3.6** |
| GCN-5-TOGL-1 | **63.0±0.8** | **76.0±3.2** | 76.0±3.2 |
| GCN-8 | 52.9±0.9 | 70.7±4.5 | 75.4±2.4 |
| GCN-7-TOGL-1 | *61.6±0.5* | *72.1±7.3* | *75.6±3.5* |
| GIN-6 | 55.7±2.8 | *73.1±3.7* | *74.1±3.1* |
| GIN-5-TOGL-1 | *61.5±1.2* | 72.7±4.2 | 73.8±3.3 |
| GIN-8 | 56.0±3.5 | 71.6±5.8 | *74.8±3.2* |
| GIN-7-TOGL-1 | *61.1±0.8* | *72.4±4.6* | 74.3±2.5 |

