# OpenReview forum: "Topological Graph Neural Networks"
_ICLR.cc/2022/Conference — ICLR 2022 Poster_

### Official Review · Reviewer_vj7C · 2021-10-29

**Correctness:** 4
**Technical Novelty And Significance:** 3
**Empirical Novelty And Significance:** 2
**Recommendation:** 6
**Confidence:** 3

**Main Review:**

# Strengths
- TOGL may be a useful tool for the GNN community, while also opening the door for some further research in Topological Data Analysis (TDA).
- The paper is well-written.

# Weaknesses
- The flow of the paper may be hard to follow for the non-GNN-expert ready (i.e. TDA-side reader) due to the lack of a short background section dedicated to the topic.
- Numerical results, though fairly solids, are not striking either.

# Minor comments :
- I think that writing $f_\theta$ instead of $f$ would improve clarity of some claims throughout the paper (it feels weird to me to write say "the map $\theta \mapsto \Psi(\mathrm{ph}(G,f))$ is differentiable", as the right-hand-side does not (visually) depends on $\theta$).

**Summary Of The Paper:**

This paper introduces TOGL, a new layer for Graph Neural Networks (GNN), making the GNN "aware" of topological information during this training phase. It differs from the closely related work *Graph Filtration Learning (GFL)* (although taking inspiration from it) as GFL is mostly a readout function (roughly, final layer in a GNN) while TOGL is a more general type of GNN-layer. Numerical experiments show how, when topological information is relevant, TOGL helps to leverage it.

**Summary Of The Review:**

This is a competent paper that builds upon existing works and provides an interesting application case of TDA and may be a good addition to GNN toolboxes.

---

> ### Author Response · Authors · 2021-11-16
> **Answers to reviewer’s vj7C comments**
>
>
>
> We first want to thank the reviewer for his positive feedback. Below we provide answers that we hope will address the reviewer’s comments. Please let us know if you wish any extra clarification.
>
> ### About introducing GNN concepts
> > The flow of the paper may be hard to follow for the non-GNN-expert ready (i.e. TDA-side reader) due to the lack of a short background section dedicated to the topic.
>
> Thank you for the feedback, we hope you understand that providing a comprehensive introduction to both computational topology and graph neural networks is extremely challenging given the space constraints of submission to ICLR. We deliberately concentrated on introducing concepts from topology as we assumed these to be less wide spread and thus access to them much more difficult. As we already reached the space limit we included a reference to a comprehensive introduction to GNNs in Section 1.
>
> ### About the numerical results
> > Numerical results, though fairly solids, are not striking either.
>
> The results from our experiments section allow us to answer the main research question of the paper, namely figuring out whether endowing graph neural networks with a topology based layer (using persistent homology) can result in better performance. Based on results from Figure 1, 3 and Table 1, we think we can answer this research question positively. This is further confirmed by the number of datasets we use. Yet, if topological information is less important for the specific task (for instance when the node features already carry most of the useful information), TOGL naturally provides less of an advantage (Table 2), which is expected.
>
> What is more, we also want to highlight the extensive experimental setup. We study the impact of the position of the topological layer (Table S7), the type of embedding function (Table S8) and the total number of layers (Table S9). We hope that this provides the readers with a comprehensive study to assess the relevance of a topological based layer in graph neural networks.
>
> ### About a more explicit notation for the dependency of the function on weights
>
> > I think that writing fθ instead of f would improve clarity of some claims throughout the paper (it feels weird to me to write say "the map θ↦Ψ(ph(G,f)) is differentiable", as the right-hand-side does not (visually) depends on θ).
>
> We agree with the reviewer that this suggestion improves clarity and we updated the text accordingly.

---

> > ### Comment · Reviewer_vj7C · 2021-11-22
> > **Thanks**
> >
> > Thanks for taking time answering my review.

---

> > > ### Author Response · Authors · 2021-11-22
> > > **re: thanks**
> > >
> > > Thanks for acknowledging our changes! Since today is the last day of the discussion phase, please let us know if there's anything else we can clarify for you! Please kindly consider updating your score in case you find our answers to be satisfactory.

---

### Official Review · Reviewer_PtXE · 2021-10-31

**Correctness:** 3
**Technical Novelty And Significance:** 3
**Empirical Novelty And Significance:** 2
**Recommendation:** 6
**Confidence:** 3

**Main Review:**

The paper is well-motivated, well-written, and builds on a good theoretical framework. The authors also explicitly mention the limitations that they had to impose due to computational constraints (such as l1 constraints). They also show that in practice even using limited topological features such as connected components and cycles, helps GNNs perform better. There are three questions that I would like to ask the authors:

1- Diffusion-based models also capture some global topological information about the graphs and it has been shown that using them in both supervised and contrastive manner significantly help GNNs. How diffusion-based models are compared to your model and what if there are any theoretical links between them and your work.

2- I am curious to see how your model can contribute to contrastive learning with linear evaluation protocol as it seems it can introduce a good amount of signal.

3- I was hoping to see this method to be applied to large-scale benchmarks on OGB rather than small (and almost overfitted) TUdataets. Benchmarking the model on those datasets can show the scalability and the performance of the proposed model.

4- Finally, expanding more on the theoretical aspects of the work and maybe providing examples would make the paper more eadible.

**Summary Of The Paper:**

Authors propose a topology-aware layer that is compatible with GNNs and can encode connected components and cycles to let the GNN learn better representations based on such global topological features.

**Summary Of The Review:**

The paper roots in a nice theoretical framework but shorts fall on experimental side.

---

> ### Author Response · Authors · 2021-11-16
> **Answers to reviewer’s PtXE comments (part 2)**
>
> ### About adding more theoretical details and examples
> >Finally, expanding more on the theoretical aspects of the work and maybe providing examples would make the paper more edible.
>
> We are aware that our work builds upon a large basis of previous theoretical work. This unfortunately makes it impossible for us to include a comprehensive introduction to the topics at hand in the main text such that we had to limit the amount of explanations to one page. Nevertheless, we introduce all concepts required for the paper (i.e. simplicial and persistent homology and graph filtrations) in Appendix A of the original submission. This section also contains additional illustrative examples to support the introduced concepts.
>
> Furthermore, the submission contains detailed examples on the expressivity of WL on regular graphs in Appendix C.
>
>
> ### About the apparent shortcomings of the experimental side
> > The paper roots in a nice theoretical framework but falls short on the experimental side.
>
> We thank the reviewer for this positive feedback on our theoretical framework. However, we beg to differ regarding the experimental setup. The results from our experiments section allow us to answer the main research question of the paper, namely figuring out whether endowing graph neural networks with a topology based layer (using persistent homology) can result in better performance. Based on results from Figure 1, 3 and Table 1, we are convinced we can answer this research question positively. This is further confirmed by the number of datasets we use. Yet, if topological information is less important for the specific task (for instance when the node features already carry most of the useful information), TOGL naturally provides less of an advantage (Table 2), which is expected.
>
> What is more, we also want to highlight the extensive experimental setup. We study the impact of the position of the topological layer (Table S7), the type of embedding function (Table S8) and the total number of layers (Table S9). We hope that this provides the readers with a comprehensive set of empirical results to assess the relevance of a topology-based layer in graph neural networks.

---

> ### Author Response · Authors · 2021-11-22
> **Any additional clarification ?**
>
> Dear Reviewer,
>
> Thanks again for your feedback,
>
> Since the discussion period ends today, we were wondering if our answers manage to address your comments ?
>
> We would be happy to provide any further clarification.
>
> Kind Regards,

---

### Official Review · Reviewer_CsQm · 2021-11-03

**Correctness:** 3
**Technical Novelty And Significance:** 3
**Empirical Novelty And Significance:** 2
**Recommendation:** 6
**Confidence:** 3

**Main Review:**

The paper provides new insights into topological aware GNN and I summarize the pros as follows:
1. The work is well motivated. First of all, Figure 1 gives a motivational toy example to demonstrate the necessity to capture the topological structure. I really like the example, which is clear and easy to understand. Besides, I appreciate that the authors provide a comprehensive introduction and research roadmap in the related work section.

2. The work is technically sound and relatively novel as it introduces persistent homology, which is more powerful than the WL-test from the view of topological awareness (Theorem 2), and provides a reasonable framework to integrate it.

Comments:
1. I am confused about the topological structure awareness power of GCN illustrated in Figure 1. From both two subfigures, Vanilla GCN overperforms WL-test. The results are a conflict with the analysis of GIN~[1], where GCN should be less powerful. From my perspective, GCN first aggregates the neighbourhood features and then pool features with a mean operator. Such a message passing mechanism should be less powerful than WL-test.

2. There may be a gap between the Betti numbers and the learned graph signal from the vertex-based filtrations, as no specific training loss was used to guarantee the alignment between the two terms.

3. The author used the DeepSets embedding function as in section 4.1 to embed different graph signals into high-dimension space. However, the paper contains no empirical study to validate the choice.

4. Though TOGL is demonstrated strictly powerful than WL-test, the experiment results from Table 2 showing that WL-test overperforms in several datasets.

[1] Xu, K., Hu, W., Leskovec, J. & Jegelka, S. How Powerful are Graph Neural Networks? Arxiv (2018).

**Summary Of The Paper:**

Motivated from the field of topological data analysis (TDA), the paper proposed a Topological Graph Layer (TOGL) plugin for Graph Neural Networks to boost the ability of topological structure detection.

**Summary Of The Review:**

Motivated by topological data analysis, the paper proposed the TOGL plugin for GNN to boost the ability of topological structure detection. The idea is insightful and is analyzed theoretically and empirically. Though I have a few concerns as listed in Main Review, I would like to weakly accept the work. Surely, I will appreciate it if any of my concerns can be addressed.

---

> ### Author Response · Authors · 2021-11-22
> **Additional clarification needed ?**
>
> Dear Reviewer,
>
> Thanks again for your feedback,
>
> Since the discussion period ends today, we were wondering if our answers manage to address your comments ?
>
> We would be happy to provide any further clarification.
>
> Kind Regards,

---

### Official Review · Reviewer_2oqu · 2021-11-03

**Correctness:** 3
**Technical Novelty And Significance:** 3
**Empirical Novelty And Significance:** 3
**Recommendation:** 6
**Confidence:** 3

**Main Review:**

The paper is well motivated and mostly clear and easy to follow.
The method is well formulated and the experiments details are clear.

In terms of weaknesses of the paper, I have two main concerns:

1. Indeed, as discussed, the aspect of oversmoothing is important in the case of GNNs. However, for a better verification of the existence/absence of oversmoothing with the proposed method, much layers need to be stacked, and preferably more datasets (like Cora, Citeseer and Pubmed) ,should be compared against to other recent methods like GCNII [Chen et al. 2020] .

2.In the context of topology, which is the main contribution and scope of this work, it would be interesting to see more geometrically oriented data sets like ABC or Thingi10k (this is just an example), where objects with different topologies are given. Also, it would be interesting if the authors can discuss a recent ICCV paper that deals with geometrical data using persistent homology :
"Persistent Homology based Graph Convolution Network for Fine-grained 3D
Shape Segmentation"


**Summary Of The Paper:**

The authors present a topology analysis improvement to GCN, using persistent homology, to capture global information regarding the topology of the graph.
The authors conduct several experiments, from graph to node classification, and also introduce two novel data sets to exemplify the importance of topology.
In most cases, the proposed method outperforms other baseline methods, as well as other topology aware methods.



**Summary Of The Review:**

The paper is clearly written and adds a contribution to the GNNs society. Also, the numerical experiments suggest improvement over other data sets. Some of the experiments can be improved, and since topology is of high interest geometrical datasets and application, I think that some discussion should be added, and will strengthen the paper.

---

### Author Response · Authors · 2021-11-29
**Any more comments ?**

Dear Reviewers,

Thank again for your insightful and encouraging feedback. As the discussion period is about to end, we would like to kindly ask you if there is anything else that you would like us to clarify ?

As a reminder about the changes we brought to address your initial concerns:

- We added an extra experiment on geometric datasets where we shown that our layer clearly outperforms the baselines.
- We added a large scale dataset from OGB to our experiments that further confirms our initial results.
- We implemented an extra experiment to investigate the impact of the number of layers (and of over-smoothing) on our approach. We showed that TOGL was still providing a significant advantage over the baselines when the number of layers becomes large.

If you are satisified with our answers, please consider reflecting it in your score :)

---

### Decision · Program_Chairs · 2022-01-20

**Decision:**

Accept (Poster)

**Comment:**

This paper presents a new graph neural network layer that is sensitive to topological structure in the graph. Reviewers all believe the work is technically sound, and the experiments (particularly after author revisions) show clear benefits in cases where topological structure is important. The main questions are about whether the experimental evaluation is sufficient. While there are always more experiments that could be run, I tend to agree with the authors that the chosen experiments support the key claims in the paper, so it seems ok. The other question about the experiments is if they sufficiently convince the reader that topological structure is useful in practice. This seems more mixed. The paper would certainly be improved if there was a motivating application where there was a clear win. For example, molecular structures are used as motivation in the intro, but the best performing method on proteins doesn’t use the topological layer. All-in-all, though, there does appear to be clear improvements on carefully constructed cases, and there appear to be some benefits in real-world datasets.